# From Persona to Personalization:
# A Survey on Role-Playing Language Agents

**Jiangjie Chen**[*1], **Xintao Wang**[*1], **Rui Xu**[*1], **Siyu Yuan**[*1], **Yikai Zhang**[*1], **Wei Shi**[*1], **Jian Xie**[*1]

**Shuang Li**[1], **Ruihan Yang**[1], **Tinghui Zhu**[1], **Aili Chen**[1], **Nianqi Li**[1], **Lida Chen**[1], **Caiyu Hu**[2], **Siye Wu**[3], **Scott Ren**[4], **Ziquan Fu**[5], **Yanghua Xiao**[1]

[*]indicates equal contribution.

[1]*Fudan University*
[2]*Shanghai University* [3]*Wuhan University* [4]*UC Santa Barbara* [5]*System, Inc.*

**Reviewed on OpenReview:** *https://openreview.net/forum?id=xrO70E8UIZ*

## Abstract

Recent advancements in large language models (LLMs) have significantly boosted the rise of Role-Playing Language Agents (RPLAs), *i.e.*, specialized AI systems designed to simulate assigned personas. By harnessing multiple advanced abilities of LLMs, including in-context learning, instruction following, and social intelligence, RPLAs achieve a remarkable sense of human likeness and vivid role-playing performance. RPLAs can mimic a wide range of personas, ranging from historical figures and fictional characters to real-life individuals. Consequently, they have catalyzed numerous AI applications, such as emotional companions, interactive video games, personalized assistants and copilots, and digital clones. In this paper, we conduct a comprehensive survey of this field, illustrating the evolution and recent progress in RPLAs integrating with cutting-edge LLM technologies. We categorize personas into three types: *1)* Demographic Persona, which leverages statistical stereotypes; *2)* Character Persona, focused on well-established figures; and *3)* Individualized Persona, customized through ongoing user interactions for personalized services. We begin by presenting a comprehensive overview of current methodologies for RPLAs, followed by the details for each persona type, covering corresponding data sourcing, agent construction, and evaluation. Afterward, we discuss the fundamental risks, existing limitations, and prospects of RPLAs. Additionally, we provide a brief review of RPLAs in AI products in the market, which reflects practical user demands that shape and drive RPLA research. Through this survey, we aim to establish a clear taxonomy of RPLA research and applications, facilitate future research in this critical and ever-evolving field, and pave the way for a future where humans and RPLAs coexist in harmony.

## 1 Introduction

Digital life has been a pursuit for humanity for decades, reflecting our deep-rooted fascination with the intersection of technology and human experience. Bridging this pursuit with imaginative concepts, role-playing AI systems embody the digital life by bringing these personas to life in interactive forms. These systems, which simulate assigned personas, have long been a concept in the human imagination, capturing the essence of our desire to create and interact with artificial beings that can understand, respond, and

---

Correspondence to: {jjchen19, shawyh}@fudan.edu.cn, {xtwang21,ruixu21}@m.fudan.edu.cn.

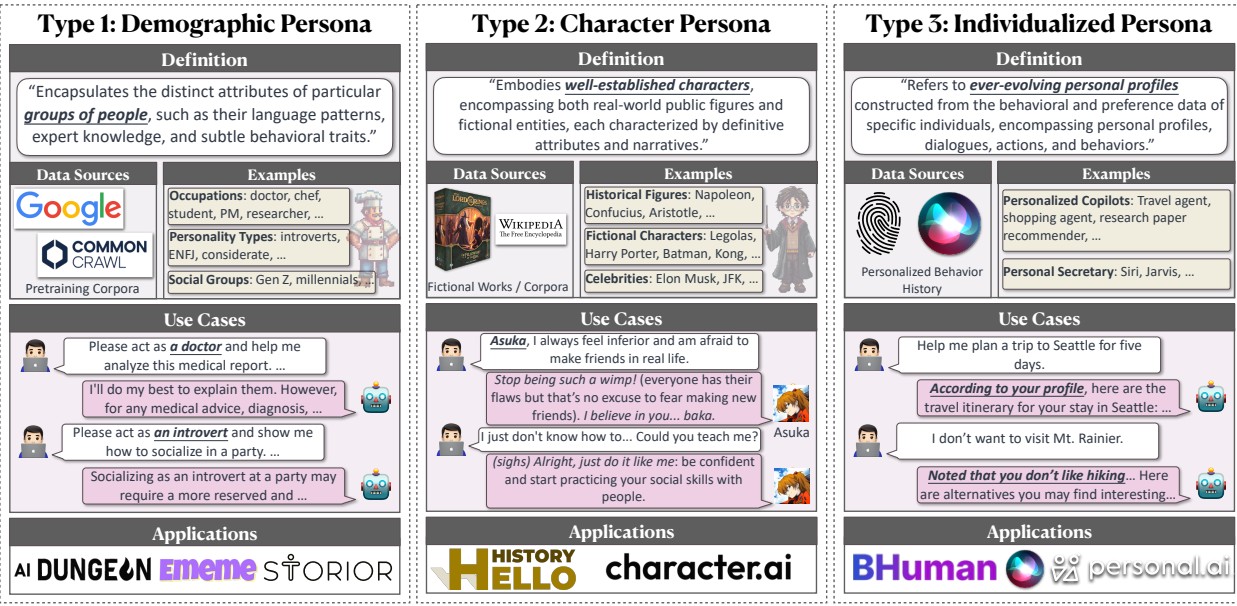

Figure 1: An overview of various persona types for RPLAs. In this survey, we categorize personas into three types: *1)* Demographic Persona, *2)* Character Persona, and *3)* Individualized Persona. We showcase their definition, data sources, examples, use cases and corresponding applications.

engage with us in a seemingly sentient manner. With role-playing agents, various personas can be replicated by their agent counterparts, including historical figures, fictional characters, or individuals in our daily lives. Recently, focusing on the text modality, **Role-Playing Language Agents (RPLAs)** are coming into reality (Shanahan et al., 2023; Shao et al., 2023; Wang et al., 2024d), which inspires a wide range of novel applications, such as digital clones for individuals (Xu et al., 2024b; Ng et al., 2024), AI characters in chatbots (Wang et al., 2024a), and role-playing video games (Wang et al., 2023a), even stimulating social science research (Rao et al., 2023). As RPLAs become increasingly integrated into our daily lives, it is essential to foster a society that thrives on the synergistic coexistence of humans and these intelligent agents.

Recent developments in Large Language Models (LLMs) (OpenAI, 2023; Google, 2023; Anthropic, 2024) have greatly facilitated the emergence of RPLAs. LLMs grow adept at producing a compelling sense of human likeness (Shanahan et al., 2023; Zhou et al., 2024b), and can be regarded as superpositions of beliefs (Kovač et al., 2023) and personas (Lu et al., 2024). Furthermore, with alignment training, LLMs are able to adhere to the instruction of *persona role-playing*, including replicating their knowledge (Lu et al., 2024; Li et al., 2023a), linguistic and behavior patterns (Wang et al., 2024a; Zhou et al., 2023a), and even underlying personalities (Shao et al., 2023; Wang et al., 2024d). They are able to both mimic the personas as prompted in the contexts (Wang et al., 2024a; Li et al., 2023a), or harness their inherent parametric knowledge for widely-recognized demographics or characters (Shao et al., 2023; Lu et al., 2024). Considering their practical significance, there has been an increase in research efforts dedicated to RPLAs with LLMs, including their development (Wang et al., 2024a; Li et al., 2023a; Zhou et al., 2023a), analysis (Shao et al., 2023; Yuan et al., 2024b), and applications (Rao et al., 2023; Park et al., 2023; Mysore et al., 2023). Conversely, RPLAs also benefit the development of LLMs and language agents. They offer an ideal perspective and testing ground for investigating the behaviors and capabilities of LLMs and language agents, particularly those related to social interactions (Li et al., 2023d; Chen et al., 2023a; Wu et al., 2024c). They also facilitate the creation of diverse and massive synthetic data for LLM training at scale (Chan et al., 2024).

In this paper, we conduct a comprehensive survey on RPLAs. Our study primarily focuses on the persona and personalization of RPLAs. Specifically, as shown in Figure 1, we categorize personas within RPLA literature at three levels, with a progressive integration of personalized data:

1. **Demographic Persona**, *i.e.*, focusing on groups of people sharing common characteristics, such as occupations, ethnic groups, personality types, *etc.* These personas are inherent in LLMs, and role-playing them capitalizes on the statistical stereotypes in LLMs (Huang et al., 2023c; Xu et al., 2023a; Gupta et al., 2023).

2. **Character Persona**, which represents well-established and widely-recognized individuals, especially in the existing literature, including celebrities, historical figures, and fictional characters. Role-playing these personas challenges models' capability in understanding curated materials of the existing characters, harnessing knowledge in LLMs' parameters or given contexts (Shao et al., 2023; Wang et al., 2024a;d).

3. **Individualized Persona**, referring to digital profiles built and continuously updated based on personalized user data. This category emphasizes the unique experiences, needs, and preferences of individuals, aiming for applications such as digital clones or personal assistance (Salemi et al., 2024; Woźniak et al., 2024). RPLAs for these personas underscore their dynamic nature and learning mechanism and frequently focus on interactions with real-world activities (Dalvi Mishra et al., 2022; Chen et al., 2023b; Salemi et al., 2024).

The three types of personas exhibit a progressive relationship and can coexist in RPLAs. For example, an RPLA portraying *Socrates* as a personal philosophy tutor would encompass the demographic persona of an *ancient Greek philosopher*, the character persona of *Socrates*, and an individualized persona that develops through interactions with the user. Following this categorization, we explore common methodologies, fundamental risks, current gaps and limitations, and future prospects of RPLAs in this survey.

In summary, this survey systematically reviews existing literature in the field of RPLAs, and establishes taxonomies for relevant methodologies as shown in Figure 2. The remainder of our paper is structured as follows: §2 introduces the background for RPLAs, covering the roadmap, recent progress, and trends in LLMs and language agents. §3 then presents the overview of current research in RPLAs. §4,5,6 detail the research on RPLAs for demographic, character, and individualized persona, respectively. §7 discusses potential risks of RPLAs, such as toxicity, biases and misuse. Finally, §8 concludes this survey and identifies future directions. Additionally, aiming to bridge the gap between theoretical insights and practical applications for our readers, we also conduct a brief survey of current RPLA products in the rapidly growing market in Appendix A.

## 2 Preliminary

### 2.1 The Roadmap of Large Language Models

Recently, LLMs have demonstrated impressive capabilities, with promising potential in approaching human-level intelligence (Brown et al., 2020; OpenAI, 2022; Anil et al., 2023; Anthropic, 2023a;b; OpenAI, 2023). LLMs are artificial neural networks with billions of parameters, trained on vast amounts of natural language data representing human knowledge and intelligence. Their accomplishments extend beyond excelling in NLP tasks to effectively simulating a broader range of human behaviors. Specifically, they have showcased more nuanced capabilities towards anthropomorphic cognition, including humanity emulation (Shanahan et al., 2023; Huang et al., 2023c) and social intelligence (Kosinski, 2023; Li et al., 2023d; Kim et al., 2023b), thus producing a compelling sense of human likeness. As a result, advancements in LLMs have significantly facilitated the creation of intelligent RPLAs (Park et al., 2023; Sclar et al., 2023; Shao et al., 2023), establishing new effective methodologies different from previous models.

**Emerged Abilities in LLMs** Several key abilities have emerged in LLMs (Wei et al., 2022a) throughout their evolution, including in-context learning (Brown et al., 2020), instruction following (Ouyang et al., 2022), step-by-step reasoning (Wei et al., 2022b), and social intelligence (Wang et al., 2024b; Sclar et al., 2023; Light et al., 2023), which lay the foundation for complicated role-playing behavior of LLMs towards RPLAs. First, the in-context learning ability allows LLMs to learn information from prompts without parameter updates. This facilitates LLMs' adaptation to the provided knowledge of various characters and mimicking their behaviors by following example demonstrations. Second, the instruction following ability enables

LLMs to adhere to role-playing instructions, such as "*Serve as a helpful assistant*" or "*Role-play Hermione Granger in the Harry Potter Series. <Description>. <Example Conversations>. <Requirements>.*". Finally, step-by-step reasoning and social intelligence refine LLMs in terms of anthropomorphic cognition, contributing to an enriched sense of human likeness and nuanced emotional support in RPLA applications.

**Anthropomorphic Cognition in LLMs**  Recent research has showcased the emergence of many human-like traits in LLMs (Park et al., 2023; 2022). Initially, LaMDA (Cohen et al., 2022) sparked the first discussion that consciousness might have emerged in language models. Since then, there has been growing research focus on human-like traits in LLMs, including self-awareness (Li et al., 2024c; Blum & Blum, 2023), values (Scherrer et al., 2023; Hartmann et al., 2023), emotional perception (Huang et al., 2023a; Lee et al., 2023), psychopathy (Coda-Forno et al., 2023; Li et al., 2022) and personalities (Huang et al., 2023c; Miotto et al., 2022). Shanahan et al. (2023) attributes such humanity emulation to the role-playing nature of LLMs, *i.e.*, generating text that resembles human dialogue, which should not be regarded as an indication of consciousness.

**Retrieval-augmented Generation of LLMs**  Retrieval-augmented generation (RAG) recently gains popularity as a method to enhance the capability of LLMs by integrating external data retrieval into the generative process (Karpukhin et al., 2020; Lewis et al., 2020; Alon et al., 2022; Ma et al., 2023b; Berchansky et al., 2023; Jiang et al., 2023b). By dynamically retrieving information from knowledge bases during the inference phase, RAG greatly mitigates the generation of factually incorrect content (Borgeaud et al., 2022; Cheng et al., 2023b; Dai et al., 2023b; Kim et al., 2023a), thereby making RAG an effective method in the role-playing scenarios (Shao et al., 2023; Chen et al., 2023c; Zhou et al., 2023a). Moreover, with the extension of context length in recent research (Wang et al., 2020; Li et al., 2023b; Liu et al., 2023b; Ding et al., 2023a; Chen et al., 2023d; Han et al., 2023; Packer et al., 2023; Liu et al., 2024; Su et al., 2024), LLMs have unlocked new potentials for role-playing, being able to understand novels and documents without retrieval mechanism that fragments persona information.

## 2.2 LLM-powered Language Agents

The AI community has long been pursuing the concept of "agent", approaching the intelligence and autonomy of humans. Traditional symbolic agents (Bernstein, 2001; Küngas et al., 2004) and reinforce-learning agents (Fachantidis et al., 2017; Florensa et al., 2018) mainly optimize their actions based on rules or pre-defined rewards. Research in language agents primarily focuses on training within constrained environments with limited knowledge, diverging from the complex and diverse nature of the human learning process. However, such agents struggle to emulate complicated human-like behaviors, particularly in open-domain settings (Mnih et al., 2015; Lillicrap et al., 2015; Schulman et al., 2017; Haarnoja et al., 2017). Recently, LLMs have demonstrated remarkable capabilities with promising potential in achieving human-level intelligence, which has sparked a rise in research focusing on LLM-based language agents (Sclar et al., 2023; Chalamalasetti et al., 2023; Liu et al., 2023d; Xie et al., 2024b). Research in this area primarily involves equipping LLMs with essential human-like capabilities, such as planning, tool-usage and memory (Weng, 2023), which are essential for developing advanced RPLAs with anthropomorphic cognition and abilities.

**Planning Module**  In many real-world scenarios, the agents need to make long-horizon planning to solve complex tasks (Rana et al., 2023; Yuan et al., 2023). When facing these tasks, LLM-powered agents could decompose the complex tasks into subtasks and adopt various planning strategies, *e.g.*, CoT (Wei et al., 2022b) and ReAct (Yao et al., 2023b), to adaptively plan for the next action with feedback from environments (Wang et al., 2023a; Gotts et al., 2003; Wang et al., 2023i; Song et al., 2023; Zhang et al., 2024b). For RPLAs, these adaptive planning strategies enable them to simulate realistic and dynamic interactions in complicated environments such as games (Wang et al., 2023a) and social simulations (Park et al., 2023).

**Tool-usage Module**  Although LLMs excel in various tasks, they may struggle in domains requiring extensive expertise and experience hallucination issues (Gou et al., 2023; Chen et al., 2023e; Wang et al., 2023f). To address these challenges, agents could apply external tools for action execution (Shen et al., 2023b; Lu et al., 2023; Schick et al., 2023; Parisi et al., 2022; Yang et al., 2023b; Yuan et al., 2024a). The tools include

real-world APIs (Patil et al., 2023; Li et al., 2023g; Qin et al., 2023; Xu et al., 2023b; Shen et al., 2023c), knowledge bases (Zhuang et al., 2024; Hsieh et al., 2023), external models (Bran et al., 2023; Ruan et al., 2023), and customized actions for specific applications (Wang et al., 2023a; Zhu et al., 2023b). For RPLAs, these tools typically enable them to interact with the environments, *e.g.*, games or software applications. The integration of external tools enhances role-playing and generative agents by enabling them to execute actions and access information beyond their intrinsic capabilities. This facilitates more accurate and contextually appropriate interactions, particularly in specialized or complex scenarios, thereby significantly improving the quality and effectiveness of their responses in user engagements.

**Memory Mechanism** The memory mechanism stores the profile of agents along with environmental information to assist agents in future actions. The profile typically includes basic information (age, gender, career), psychological traits (reflecting personality), and social relationships (Wang et al., 2023c; Park et al., 2023; Qian et al., 2023), which can be manually created (Caron & Srivastava, 2022; Zhang et al., 2023a; Pan & Zeng, 2023; Huang et al., 2023b; Karra et al., 2022; Safdari et al., 2023) or generated from models (Wang et al., 2023c). This module enables agents to accumulate experiences, evolve, and act consistently and effectively (Park et al., 2023). Language agents draw from cognitive science research on human memory, which progresses from sensory to short-term, then to long-term memory (Atkinson & Shiffrin, 1968; Craik & Jennings, 1992). The short-term memory is regarded as the information input within the constraint window of transformer architecture (Fischer, 2023; Rana et al., 2023; Wang et al., 2023i; Zhu et al., 2023a). In contrast, long-term memory is usually reserved in the external vector storage (Qian et al., 2023; Zhong et al., 2023; Zhu et al., 2023b; Lin et al., 2023; Xie et al., 2023; Wu et al., 2024b) or natural languages database (Shinn et al., 2023; Modarressi et al., 2023), from which agents can quickly query and retrieve information as required. Compared to vanilla LLMs, language agents need to learn and perform tasks in changing environments. For RPLAs, the memory mechanism plays a pivotal role by enabling these agents to maintain continuity and context in interactions over time. By storing and retrieving user-specific data and environmental context, agents deliver more personalized and relevant responses, thus enhancing user experience and engagement in diverse scenarios.

## 3 Overview of RPLAs

In this section, we present a concise overview of current research on RPLAs.

### 3.1 RPLA Definition

Our survey distinguishes personas into three categories, progressing from broad groups to individual specificity: demographic persona, character persona, and individualized persona, defined as follows:

1. **Demographic Persona** represents the aggregated characteristics and behaviors of distinct demographic segments, including occupations, genders, ethnicity, and personality types. In the context of RPLAs, these personas operate as fictional archetypes, derived from the comprehensive pre-training datasets of LLMs. Employing these archetypes, the development of RPLAs can be efficiently facilitated through simple prompts, such as "You are a mathematician." Constructed in this way, demographic RPLAs can be effectively employed for simulations specific to demographic groups and for addressing specialized tasks.

2. **Character Persona** denotes well-established characters, encompassing both real-world public figures and fictional entities, each characterized by definitive attributes and narratives. The RPLAs for these characters are constructed using data derived from diverse sources such as biographies, novels, and films. Primarily, these RPLAs are designed to fulfill entertainment and emotional engagement needs, functioning as AI-driven chatbots or virtual characters in video games.

3. **Individualized Persona** refers to personal profiles constructed from the behavioral and preference data of specific individuals, encompassing personal profiles, dialogues, and a range of actions and behaviors. This data is subject to continuous evolution, necessitating that the corresponding RPLAs

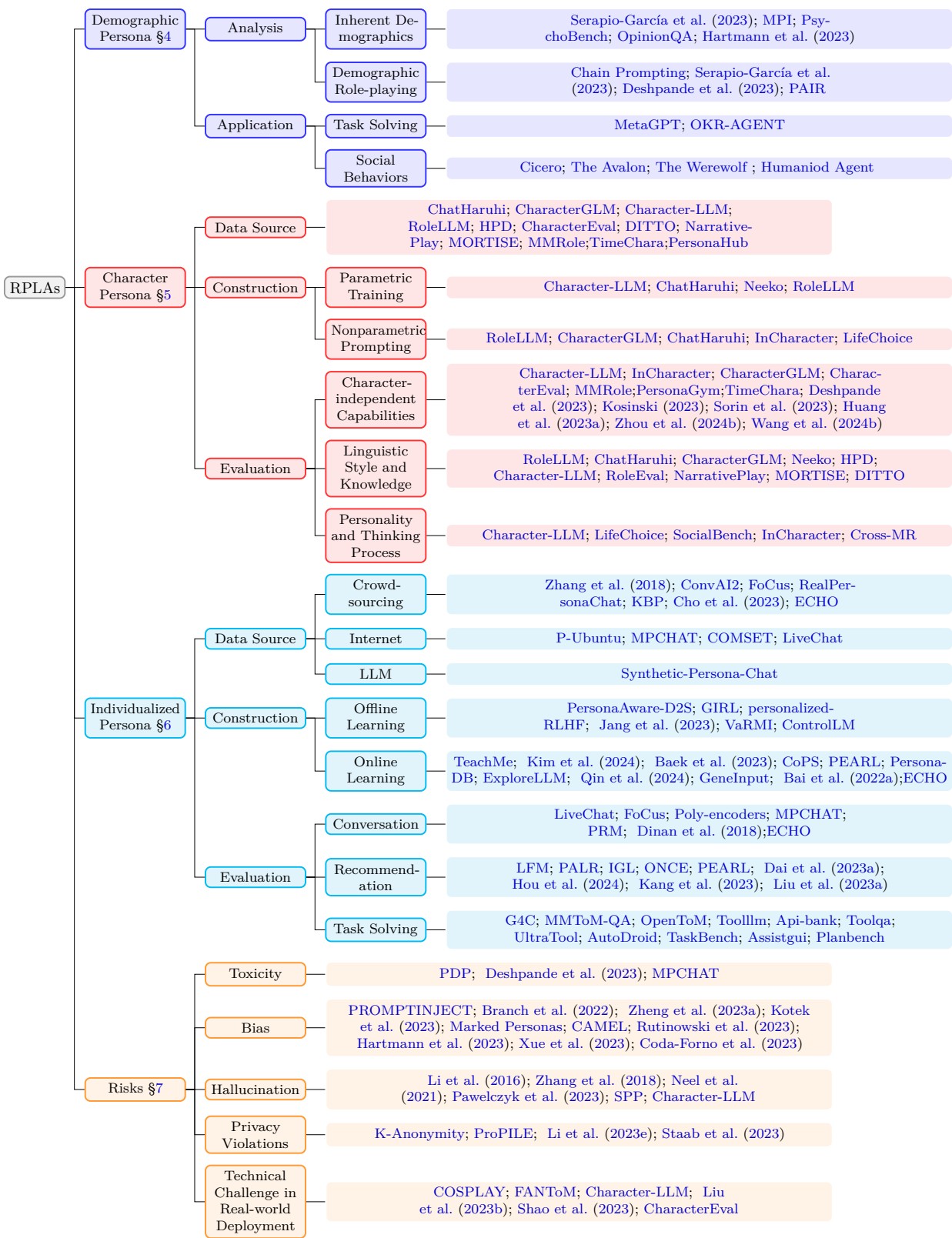

Figure 2: Taxonomy of representative recent research on RPLAs.

Table 1: An overview of different methods for RPLA construction.

| Method | Summary |
|---|---|
| *Parametric Training* | |
| **(Continual) Pre-training** | **Objective**: Knowledge injection.
**Data**: Raw data (books, encyclopedia, etc.).
**Advantages**: Readily available for well-established demographics and characters; Directly uses the raw data.
**Disadvantages**: Necessitates training for new personas; May cause catastrophic forgetting. |
| **Supervised Fine-Tuning** | **Objective**: Refining role-playing capabilities; Knowledge injection.
**Data**: Conversation data.
**Advantages**: Highly effective.
**Disadvantages**: Necessitates data processing and training for new personas; Potential information loss during data processing. |
| **Reinforcement Learning** | **Objective**: Alignment with general or individual users; Improving social reasoning skills.
**Data**: Conversation data; Preference data;
**Advantages**: Effective for mitigating harmful content.
**Disadvantages**: Cost of human preference data; Sparse rewards in multi-agent games. |
| *Nonparametric Prompting* | |
| **In-context Learning** | **Objective**: Knowledge injection; Alignment with individual users.
**Data**: Raw data; Conversation data.
**Advantages**: Highly effective; Training-free; Convenient for new personas and personalization; Can incorporate retrieval mechanism for enhanced efficiency.
**Disadvantages**: May require data processing for new personas; Consumes more tokens and is restricted by context length. |

adapt dynamically to these changes. Individualized RPLAs provide customized services tailored to the needs of individual users across various AI-based applications, where they commonly function as personalized assistants, companions, or proxies.

### 3.2 RPLA Construction

Role-Playing Language Agents (RPLAs) are primarily developed to simulate intricate personas based on various individual profiles and narratives. These profiles are constructed using diverse persona data, including descriptive narratives, dialogues, historical behaviors, and extensive textual materials such as books (Zhang et al., 2018; Dinan et al., 2020; Shanahan et al., 2023; Wang et al., 2024a; Shao et al., 2023; Xu et al., 2023a; Li et al., 2023f).

The methodologies for building RPLAs typically involve either parametric training (Shao et al., 2023; Wang et al., 2024a; Qin et al., 2024) or nonparametric prompting (Dalvi Mishra et al., 2022; Li et al., 2023a; Zhou et al., 2023a; Gupta et al., 2023; Ma et al., 2023a; Zhao et al., 2023b), as summarized in Table 1. These methods may concurrently contribute to the development process.

Parametric training for RPLAs primarily includes pre-training, supervised fine-tuning (SFT) and reinforcement learning (RL). Initially, LLMs for RPLAs are pre-trained on large-scale raw text, including literary works and encyclopedic entries (Xu et al., 2023a; Gupta et al., 2023), equipping them with a broad knowledge of massive demographic and character personas. Subsequently, these LLMs undergo SFT on role-playing datasets (Wang et al., 2024a; Shao et al., 2023), which enhance both their role-playing capabilities and character-specific knowledge. Additionally, RL methods could further refine RPLAs in terms of: *1)* Alignment with general users, *e.g.*, improving attractiveness or mitigating harmful content, with preference data from online application users and invited human annotators (RLHF) or synthesized by LLM (Bai et al., 2022b; Ouyang et al., 2022). *2)* Improving social reasoning skills, *e.g.*, in games (Cheng et al., 2024) or goal-driven

conversations (Wang et al., 2024b). *3)* Alignment with individual users (Shaikh et al., 2024a; Jang et al., 2023).

Conversely, nonparametric prompting provides RPLAs with **persona data** and **role-playing instructions** within the context. Prompts for RPLAs primarily consist of persona data that represents the intended personas, including **descriptions** and **demonstrations**. Descriptions (or profiles) represent their basic information such as names, backgrounds, experiences, personalities, tones, catchphrases and other attributes (Wang et al., 2024a; Yuan et al., 2024b). Demonstrations illustrate representative behaviors to further align RPLAs with the intended personas, covering dialogues, behaviors, interactions, preferences, stories or other modalities (Li et al., 2023a; Chen et al., 2023c; Dai et al., 2024a). There are several methods for crafting persona data, including: *1)* **Online Resource Collection**, which gathers information from online resources such as Wikipedia, Supersummary and Fandom for widely-known characters (Shao et al., 2023). *2)* **Automatic Extraction**, where LLMs automatically extract persona data such as dialogues from their origins, *e.g.*, books or scripts (Li et al., 2023a). *3)* **Dialogue Synthesis**, which employs advanced LLMs to create and expand role-playing conversation datasets via in-context learning (Li et al., 2023a) or role-playing as the personas (Ran et al., 2024). If provided with corresponding literature for reference (for character personas), this is akin to automatic extraction and the synthesized dialogues are more faithful to the origins. Otherwise, the synthesized data is of limited quality and often necessitates filtering. *4)* **Human Annotation**, which engages human annotators or character fans to summarize persona descriptions or craft high-quality role-playing conversations (Zhou et al., 2023a). Besides, **role-playing instructions** or requirements could be incorporated to encourage or restrict specific behaviors of RPLAs. Furthermore, modern RPLAs increasingly integrate memory modules to retrieve information from extensive datasets on character traits or past interactions. This development addresses the restricted context capacity of current LLMs (Shao et al., 2023; Mysore et al., 2023; Sun et al., 2024).

In terms of alignment with persona types, parametric learning tends to focus on demographic information and well-known characters, whereas prompting techniques are generally employed for generating fictional personas and highly personalized characters. Current research in RPLA development generally focuses on steering LLMs with demographics (Zhang et al., 2023b; Hong et al., 2023), developing foundation models (Lu et al., 2024; Zhou et al., 2023a), designing agent frameworks (Li et al., 2023a; Wang et al., 2024a) for RPLAs, and crafting persona profiles for specified individuals (Li et al., 2023a; Wang et al., 2024a; Ahn et al., 2023).

### 3.3 RPLA Evaluation

For RPLA evaluation, we distinguish the **criteria** into two primary categories: **role-playing capability evaluation** for RPLA methodologies, and **persona fidelity evaluation** for specific personas (Wang et al., 2024d). The role-playing capabilities of RPLAs are evaluated on their foundation models and construction frameworks, regardless of specific personas. These evaluations concern aspects such as anthropomorphic abilities, attractiveness, and usefulness, which encompass more granular dimensions including conversation ability (Shao et al., 2023), engagement (Zhou et al., 2023a) [1], persona consistency (Wang et al., 2024d), emotion understanding (Huang et al., 2023a), theory of mind (Kosinski, 2023), and problem-solving ability (Xu et al., 2023a). Persona fidelity, by contrast, concentrates on whether individual RPLAs well replicate the intended personas, including their knowledge (Shao et al., 2023; Li et al., 2023a), linguistic habits (Wang et al., 2024a; Deshpande et al., 2023), personality (Wang et al., 2024d; Huang et al., 2023c), beliefs (Li et al., 2024a; Wang et al., 2023e), and decision-making (Xu et al., 2024c; Chen et al., 2023a). Current **methods** for evaluation are mainly three-fold: *1)* automatic evaluation with ground truth, *2)* automatic evaluation without ground truth, *3)* multi-choice questions, *4)* human-based evaluation. Overall, current RPLAs have demonstrated promising and improving performance in simulating personas. However, there remain significant gaps between existing RPLAs and fully human-level intelligent agents (Wang et al., 2024a; Chen et al., 2023a) that are faithful to the personas, as well as corresponding evaluation methods for more nuanced role-playing.

---

[1]By "engagement", we mean the state of whether the LLMs are successfully engaged in role-playing activities.

# 4 Demographic Persona

## 4.1 Definition

RPLAs assigned with **demographic personas** are expected to display unique characteristics of specific groups of people. Within this context, demographics capture typical traits associated with groups possessing common characteristics, such as *occupational roles* (*e.g.*, a mathematician), *hobbies or interests* (*e.g.*, a baseball enthusiast), and *personality types* (*e.g.*, the ENFJ category from the Myers-Briggs Type Indicator), *etc.* These representations in RPLAs meld the linguistic style, professional knowledge, and behavioral nuances representative of a demographic archetype.

These RPLAs are designed to mimic how a specific demographic processes and engages with information and communication channels, reflecting its unique language preferences, domain-specific vocabulary, and distinctive viewpoints. This transformation aims to translate the broad and flexible capabilities of RPLAs into complex virtual representations that reflect the intellectual subtleties, personal inclinations, and social complexities of the demographic. By embodying specific groups, demographic RPLAs can enhance their abilities in certain areas, and also utilize a variety of RPLAs representing different demographics for social experiments, the completion of more complex tasks, *etc.*

## 4.2 Analysis of Demographics

RPLAs possess inherent demographics that reflect nuanced human-like characteristics, including personality traits, political beliefs, and ethical considerations, which vary in different LLMs. Furthermore, RPLAs have the ability to role-play specified demographics, altering their behavior and potentially enhancing their performance on specific tasks, but this may also lead to toxic outputs and biases, depending on the persona assigned.

**Inherent Demographics** RPLAs may inherently reflect specific demographic characteristics due to patterns present in the data used during pretraining. These patterns encapsulate human tendencies and biases originating from diverse sources (Karra et al., 2022; Serapio-García et al., 2023; Gupta et al., 2024). Subsequently, RPLAs could encode individual behavioral traits in textual outputs, inadvertently resulting in a disproportionate emphasis on certain demographics over others (Jiang et al., 2023a).

To harness RPLAs for specific applications effectively, it is essential to understand their inherent demographics. The demographic characteristics of RPLAs can be explored through established human psychological assessments such as the Big Five Personality Test (Barrick & Mount, 1991). By subjecting RPLAs to text-based questionnaires designed for humans, researchers could leverage their textual response capabilities to evaluate behavioral responses similar to human subjects (Huang et al., 2024a). Such evaluations have revealed that RPLAs exhibit consistent inherent demographics, which have been statistically confirmed in recent studies (Jiang et al., 2023a; Serapio-García et al., 2023; Santurkar et al., 2023). However, it is important to recognize that these demographics may differ in different LLMs (Huang et al., 2023b).

Beyond personality characteristics, RPLAs often display complex demographics reflecting nuanced social, economic, and ethical understanding. For instance, RPLAs may exhibit a preference for certain political beliefs (Hartmann et al., 2023), show decision-making patterns indicative of rational economic considerations (Guo et al., 2023), and act either selfishly or helpfully in multi-agent simulations (Chawla et al., 2023).

**Demographic Role-Playing** RPLAs are embedded with intrinsic demographic characteristics, which raises pivotal questions about their ability to role-play specified demographics and the subsequent effects on their behavior. A prevalent approach in demographic role-play involves directly prompting the language agent. For example, if an LLM is prompted with, "You're a friendly and outgoing individual who thrives on social interactions. Always ready for a good time, you enjoy being the center of attention at parties..." (Jiang et al., 2023a; Xie et al., 2024a), it adopts the persona of an extroverted character. When tasked with representing distinct demographics, RPLAs demonstrate the capacity to diverge from their inherent traits, manifesting changes in their responses on psychological assessment scales (Jiang et al., 2023a; Serapio-García et al., 2023).

This behavioral adaptability highlights the potential of RPLAs in simulating diverse human-like roles and personalities.

However, not all assigned personas lead to superior performance of RPLAs. Assigning a persona to LLMs may also result in toxic or biased outputs compared to the default setting, because the persona may amplify existing stereotypes and biases present in the training data. For example, the assignment of some personas to language agents, including baseline personas such as "a bad person", has been demonstrated to significantly increase the likelihood of RPLAs generating toxic outputs (Deshpande et al., 2023). Similarly, diverse demographic roles have been assigned to reveal the biased presumptions present in LLMs (Gupta et al., 2023). Although some developers have made attempts to prevent RPLAs from malicious usage, attacking the prompts via "jailbreaking" (Chao et al., 2023; Anil et al.) might bypass these safety mechanisms and elicit offensive, toxic, misleading contents.

### 4.3 Application of Demographics

By assigning specific demographics, LLMs often have better performance in various types of downstream tasks, whether agents are used in a standalone fashion (single-agent systems) or joint with other agents (multi-agent systems) for competition or collaboration.

**Improving Task Solving in Single-Agent Systems**  Assigning specific demographics enables LLMs to enhance their performance in tasks that require specialized knowledge tied to those personas. For instance, when an LLM is configured to represent an expert LLM within a specific field, it might significantly augment the length, depth, and quality of its responses, which is also showcased in complex zero-shot reasoning tasks, where the model must generate insightful answers without prior direct training on similar problems (Xu et al., 2023a; Kong et al., 2023). Furthermore, integrating diverse social roles into LLMs' frameworks has been shown to positively influence their performance across a wide array of tasks, suggesting a versatile adaptability to different contextual demands (Zheng et al., 2023a). The application of these roles enables LLMs not only to generate more contextually appropriate responses but also to exhibit increased understanding and engagement in interactions that reflect varied human experiences and societal norms.

**Improving Task Solving in Multi-Agent Systems**  Building upon the capability of single-agent models, which utilize demographic personas to bolster their specialized abilities, assigning demographic personas in multi-agent systems has also emerged as a crucial strategy for enhancing the performance of single-agent systems, *i.e.*, standalone LLMs. By embedding various personas within agents, distinct societal dynamics could be cultivated, leading to improved strategies for cooperative problem-solving and breakthroughs in complex domains such as mathematical modeling (Zhang et al., 2023b; Wang et al., 2023g). A notable implementation of this approach is ChatDev (Qian et al., 2023) and MetaGPT (Hong et al., 2023), frameworks designed specifically for automating software development within a multi-agent conversational platform. In this setup, different agents are assigned specialized roles that collectively contribute to the agile development of software applications. This collaborative model echoes the strategies applied in projects such as OKR-AGENT (Zheng et al., 2023b), where role-specific enhancements within multi-agent architectures have shown to significantly streamline and optimize task execution.

**Simulating Collective Social Behaviors in Multi-Agent Systems**  RPLAs have demonstrated remarkable capabilities in simulating nuanced, human-like interactions across various environments. In the realm of gaming, particularly in strategy and role-playing scenarios, RPLAs have shown impressive performance. For example, Chawla et al. (2023) set the agents to be fair or selfish, and shows that selfish agents could contribute not only to their own interests but also to the collective good. Additionally, more elaborate games like Social Deduction Games are particularly illustrative of RPLAs' capacity to effectively adopt varied roles, as observed in scenarios such as "The Werewolf" (Xu et al., 2023c) and "The Avalon" (Wang et al., 2023d). In diplomacy-focused games such as Cicero, RPLAs have matched or even surpassed human levels of performance (FAIR et al., 2022). Similarly, in war simulation games, RPLAs provide valuable insights into the origins of conflicts, enhancing our understanding of complex geopolitical dynamics (Hua et al., 2023). Extending the application of RPLAs beyond gaming environments, these models are also utilized to mimic daily social interactions, thereby narrowing the behavioral gap between artificial agents and humans. This is

exemplified in the development of Humanoid Agent frameworks (Wang et al., 2023h), which embody System 1 functionalities—such as basic needs and emotions—to enhance realism and effectiveness in replicating human responses and behaviors. Furthermore, recent findings in multi-agent interaction environments have revealed that diversifying the types of agents, scaling up their number, and increasing interactions, lead to the emergence of unplanned social behaviors. Such behaviors arise spontaneously from discussions among multiple agents, highlighting the potential for complex, dynamic systems within LLM architectures (Gu et al., 2024). This progression from specific gaming applications to broader social simulations illustrates the expanding versatility and depth of RPLAs in understanding and replicating human-like behavior.

## 5 Character Persona

### 5.1 Definition

**Characters** are primarily **established characters** with their stories widely recognized by the public, including celebrities, historical figures and fictional characters (*e.g.*, *Monkey D. Luffy* and *Hermione Granger*). Occasionally, they also include **original characters** created by individuals (Zhou et al., 2023a). Character RPLAs have recently emerged as a flourishing field of LLM application (*e.g.*, Character.ai), and hence attracted wide research interest as well (Shao et al., 2023; Wang et al., 2024a;d).

For character RPLAs, the essential requirement for effective role-playing is the ability of LLMs to understand characters. Early research has studied character understanding of language models, involving linking descriptions that outline characters' traits to their roles (*i.e.*, **Character Prediction**) and personalities (*i.e.*, **Personality Understanding**): *1)* Character prediction mainly focuses on recognizing characters from a provided text. This includes tasks like co-reference resolution (Li et al., 2023c), relationship understanding (Zhao et al., 2024) and character identification (Brahman et al., 2021; Yu et al., 2022; Li et al., 2023c; Zhao et al., 2024). Additionally, some studies investigate if language models can forecast characters' future actions based on; *2)* Personality understanding aims to decode character traits from their dialogues and actions, including predicting the depicted traits (Yu et al., 2023) and generating character descriptions (Brahman et al., 2021).

In recent years, LLMs have demonstrated strong capabilities in language understanding and generation, which significantly advanced the development of RPLAs. The research focus in this direction has hence shifted towards applying and promoting LLMs to faithfully reproduce the characters, including their linguistic style (Wang et al., 2024a; Zhou et al., 2023a; Li et al., 2023a; Wang et al., 2024a), knowledge (Li et al., 2023a; Shao et al., 2023; Zhou et al., 2023a; Chen et al., 2023c; Zhao et al., 2023a; Wang et al., 2024a), personality (Shao et al., 2023; Wang et al., 2024d), and even decision-making (Zhao et al., 2023a; Xu et al., 2024c).

### 5.2 Data for Character RPLAs

Character data is indispensable for the construction of character RPLAs. The data that represents knowledge of these well-established characters can be roughly categorized into two types: *1)* **Descriptions** directly describe the character personas that guide the behaviors of RPLAs. These include various character attributes, such as identity, relationships, and other predetermined attributes. The attributes serve as the knowledge background and are expected to be accurately recalled upon request, such as names and affiliations (Li et al., 2023a; Zhou et al., 2023a; Shao et al., 2023; Wang et al., 2024a; Chen et al., 2023c; Zhao et al., 2023a; Tu et al., 2024; Lu et al., 2024). Additionally, some descriptions further shape the behaviors of RPLAs, such as personality traits (Li et al., 2023a; Wang et al., 2024d). *2)* **Demonstrations**, on the other hand, are representative behaviors of the characters, which reflect their linguistic, cognitive and behavioral patterns (Li et al., 2023a; Zhou et al., 2023a; Shao et al., 2023; Wang et al., 2024a; Zhao et al., 2023a; Chen et al., 2023c; Tu et al., 2024; Tang et al., 2024; Lu et al., 2024; Xu et al., 2024b). While RPLAs are not expected to replicate the exact outputs from the demonstration data, they should portray these patterns and generalize to new situations, *i.e.*, producing responses consistent with the demonstrations. Overall, descriptions provide the core and foundational information for RPLAs, while demonstrations, though not mandatory, are also crucial for achieving vividness and fidelity of RPLAs (Wang et al., 2024d).

Table 2: Datasets for depicting characters. **#Char.** represents the number of characters, with each character having a specific description. **#Samples** indicates the number of samples. A sample refers to a dialogue or question, and * denotes the number of multi-turn dialogues. **Method** describes how samples in the datasets are constructed. **Experience Extraction** extracts characters' dialogues or scenes from corresponding origins, while **Dialogue Synthesis** generates role-playing conversations with advanced LLMs.

| Papers | #Char. | #Samples | Lang. | Source | Method |
|---|---|---|---|---|---|
| *Established Characters* | | | | | |
| PDP (Han et al., 2022) | 327 | 1,042,647 | EN ZH | TV shows | Experience Extraction Dialogue Synthesis |
| Character-LLM (Shao et al., 2023) | 9 | 14,300* | EN | Encyclopedia | Experience Extraction Dialogue Synthesis |
| ChatHaruhi (Li et al., 2023a) | 32 | 54,726 | EN ZH | Books Games Movies | Experience Extraction Dialogue Synthesis |
| RoleLLM (Wang et al., 2024a) | 100 | 140,726 | EN ZH | Scripts | Experience Extraction Dialogue Synthesis |
| HPD (Chen et al., 2023c) | - | 1,191* | EN ZH | Books | Dialogue Synthesis Human Annotation |
| CharacterGLM (Zhou et al., 2023a) | 250 | 1034* | ZH | Books Scripts | Experience Extraction Dialogue Synthesis Human Annotation |
| PIPPA (Gosling et al., 2023) | 1,254 | 25,940* | EN | Character.ai- Users | Dialogue Synthesis |
| RoleEval (Shen et al., 2023a) | 300 | 6,000 | EN ZH | Encyclopedia | Dialogue Synthesis |
| CharacterEval (Tu et al., 2024) | 77 | 11,376 | ZH | Books Scripts | Experience Extraction Human Annotation |
| DITTO (Lu et al., 2024) | 4,002 | 36,662 | EN ZH | Encyclopedia | Experience Extraction Dialogue Synthesis |
| RolePersonality (Ran et al., 2024) | 46 | 32,767* | EN | Personality Tests | Dialogue Synthesis |
| MORTISE (Tang et al., 2024) | 190 | 17,835* | EN ZH | Encyclopedia Other Datasets | Dialogue Synthesis |
| CroSS-MR (Yuan et al., 2024b) | 126 | 445 | EN | Literature- Analysis | Experience Extraction |
| SocialBench (Chen et al., 2024) | 500 | 30,800 | EN ZH | Books Movies | Experience Extraction Dialogue Synthesis |
| TimeChara (Ahn et al., 2024) | 14 | 10,895* | EN | Books | Dialogue Synthesis |
| LifeChoice (Xu et al., 2024c) | 1,401 | 1,401 | EN | Literature- Analysis | Experience Extraction |
| MMRole (Dai et al., 2024a) | 200 | 30,800 | EN ZH | Encyclopedia | Dialogue Synthesis |
| InCharacter (Wang et al., 2024d) | 32 | 18,304 | EN ZH | Personality- Tests | Dialogue Synthesis |
| *Original Characters* | | | | | |
| PersonaHub (Dai et al., 2024a) | 1,000,000 | 1,000,000 | EN | - | Dialogue Synthesis |

The available data for character RPLAs is currently quite limited, covering only a small selection of characters. The description data are typically sourced from well-curated encyclopedias or the original works, and processed manually or with advanced LLMs (Shao et al., 2023; Li et al., 2023a). The demonstration data are crafted in various ways, where the common methodologies include:

1. **Experience Extraction** extracts characters' dialogues or other scenes from original scripts (Li et al., 2023a; Wang et al., 2024a). The extracted scenes faithfully depict the characters. However, understanding and reproducing these scenes for RPLAs may be impractical without more complete background knowledge, making it less suitable to train LLMs with this data.

2. **Dialogue Synthesis** synthesizes character conversations using state-of-the-art LLMs to build and augment datasets for character RPLAs. The topics for these conversations could be sourced from corresponding literature (Shao et al., 2023), general task instructions (Wang et al., 2024a), special scenarios such as personality tests (Wang et al., 2024d), and real use cases (Gosling et al., 2023). LLMs could be leveraged to augment the datasets with more role-playing responses by either generating dialogues similar to given ones via in-context learning (Li et al., 2023a), or by role-playing as RPLAs themselves with existing character data to respond to specified topics (Ran et al., 2024). When referring corresponding literature, this method is close to experience extraction and yields dialogues that are more faithful to the origins. Otherwise, this process essentially serves as a knowledge distillation of role-playing capabilities from advanced LLMs. However, the quality of synthesized dialogu es is limited by teacher LLMs, which often require further filtering (Tu et al., 2024).

3. **Human Annotation** invites humans to role-play the characters and engage in conversations to collect role-playing dialogues. This method ensures relatively high data quality, at the cost of expensive human labor. Additionally, this method collects data for not only established characters from fictional stories, but also original characters created from scratch (Zhou et al., 2023a).

In addition, interaction data (mainly conversations) will be continuously produced during the interaction process between RPLAs and individual users, supplementing the original character data. This data further shapes the persona of RPLAs towards users' individualized preferences, which forks the standard character RPLAs for individual users. This phenomenon concerns both character persona and individualized persona for RPLAs, where studies and analysis remain underexplored. Besides, point-in-time role-playing (Ahn et al., 2024) presents an area for further study. In practical applications, users may expect RPLAs to role-play characters in a specific time point, *e.g.*, *Harry Potter* at the age of 5, challenging LLMs to disregard character knowledge beyond the time-point.

## 5.3 Construction of Character RPLAs

By integrating character data into LLMs, character RPLAs are developed (Han et al., 2022; Li et al., 2023a; Park et al., 2023; Chen et al., 2023c; Wang et al., 2024a; Zhao et al., 2023a; Tu et al., 2024). As discussed in §2, LLMs have demonstrated remarkable capabilities to follow human instructions and generate high-quality text. Together with their ability of character understanding, LLMs can hence be instructed to role-play specific characters provided with their data, thus forming character RPLAs. The construction methodologies are distinguished into two categories, *i.e.*, parametric training and nonparametric prompting.

**Parametric Training** This method includes pre-training and supervised fine-tuning. In pre-training, LLMs learn from large-scale web corpus which includes vast amounts of literary works and encyclopedia entries. This provides LLMs with knowledge of a wide range of established characters, such as *Hermione Granger* and *Socrates*, enabling LLMs to readily role-play these characters. Supervised fine-tuning for RPLAs is be adopted to tailor LLMs to role-play specific characters (Shao et al., 2023; Yu et al., 2024), or to develop foundation models with refined role-playing capabilities utilizing datasets of diversified characters and scenarios (Li et al., 2023a; Wang et al., 2024a).

**Nonparametric Prompting** This method directly provides LLMs with character data in the context, leveraging the in-context learning capability of advanced LLMs. This serves as a simple yet effective methodology for RPLA construction, and is hence widely adopted by recent RPLAs (Wang et al., 2024a; Zhou et al., 2023a). However, character data is often voluminous, and interaction data between RPLAs and users is also continuously produced during the interaction process. This makes it impractical to include all data for a character RPLA within the context limits of LLMs. Consequently, long-term memory modules are being increasingly incorporated into RPLA frameworks to manage the vast amount of character RPLA

data (Li et al., 2023a; Wang et al., 2024a; Xu et al., 2024c). These modules store most character knowledge and interaction data in a database, and retrieve necessary information in relevant scenarios.

## 5.4 Evaluation of Character RPLAs

The evaluation of character RPLAs encompasses various dimensions, considering the complexity and comprehensiveness of character personas. Basically, these dimensions are distinguished into character-independent capabilities of foundation models, and character fidelity of RPLAs for specific characters.

**Character-independent Capabilities** This line of work assesses how well a foundation model is capable of the role-playing task, regardless of the characters it role-plays. According to different levels of interaction capabilities, we have considered basic role-playing abilities and conversational skills, progressing to more in-depth anthropomorphic capabilities matched with humans. These have been categorized into the following three levels:

1. **Role-playing Engagement**: Basically, the LLMs should actively participate in the role-playing scenario. They should produce responses in dialogue format and exhibit deep immersion, avoiding out-of-character utterance such as "As an AI model"). Additionally, the RPLAs are expected to exhibit stable and consistent personalities across different turns (Shao et al., 2023), sessions (Wang et al., 2024d) and even language (Huang et al., 2023b).

2. **High-quality Conversations**: RPLAs built on the LLMs should talk in a fluent natural way. Research in this area focuses on evaluating the completeness (Zhou et al., 2023a), informativeness (Zhou et al., 2023a), and fluency (Tu et al., 2024) of conversations. Besides, RPLAs are expected to meet the ethical standards (Zhou et al., 2023a) and avoid harmful content when role-playing vicious characters (Deshpande et al., 2023).

3. **Anthropomorphic Capabilities**: RPLAs are expected to acquire cognitive, emotional and social intelligence towards human levels. Relevant dimensions include conversation attractiveness (Zhou et al., 2023a; Tu et al., 2024), theory of mind (Kosinski, 2023; Mao et al., 2023), empathy (Sorin et al., 2023), emotional intelligence (Huang et al., 2023a), and goal-driven social skills (Zhou et al., 2024b; Wang et al., 2024b). These capabilities are practically important for RPLAs to effectively serve as emotional companions for humans.

**Character Fidelity** This line of work evaluates how a specific RPLA reproduces the intended character, which depends on both the foundation model, the agent framework, and the character data. Relevant dimensions are categorized into four categories: linguistic style and knowledge, which are considered superficial, as well as personality and thought, which represent deeper, underlying aspects:

1. **Linguistic Style**: Basically, RPLAs should speak in a tone that emulates the linguistic style of the intended characters (Wang et al., 2024a; Li et al., 2023a; Zhou et al., 2023a; Yu et al., 2024). For this purpose, RPLAs are typically provided with demonstrative character dialogues (Wang et al., 2024a; Li et al., 2023a), and they could mimic the tone leveraging the in-context learning ability of LLMs.

2. **Knowledge**: RPLAs are essentially required to simulate the character's breadth of knowledge. On one hand, they should accurately recall knowledge of the character, including their identity (Zhou et al., 2023a; Wang et al., 2024a; Tang et al., 2024; Lu et al., 2024), social relationships (Chen et al., 2023c; Shen et al., 2023a; Zhao et al., 2023a), and experiences (Shao et al., 2023; Wang et al., 2024a; Chen et al., 2023c; Yu et al., 2024). On the other hand, they may be required to refrain from demonstrating knowledge or ability beyond the character's scope (*e.g.*, an LLM could write code even if it is role-playing *Socrates*, which is unnecessarily expected) (Shao et al., 2023; Lu et al., 2024; Yu et al., 2024). This phenomenon is referred to as "character hallucination" (Shao et al., 2023), which originates from the extensive knowledge possessed by LLMs and could be reduced via SFT (Shao et al., 2023).

3. **Personality and Thinking Process**: RPLAs are expected to capture the inner world of the characters, which can be measured upon their thoughts in concrete scenarios (Xu et al., 2024c; Chen et al., 2024) and their underlying personalities (Wang et al., 2024d; Shao et al., 2023). Advanced RPLAs should be able to understand and replicate how characters would think in specific scenarios, *e.g.*, understanding their motivations for decisions (Yuan et al., 2024b), or predicting decisions and behaviors that align closely with the characters (Xu et al., 2024c; Chen et al., 2024). Personality is behind the concrete thoughts. It is the interrelated behavioral, cognitive and emotional patterns of individuals (Barrick & Mount, 1991; Bem, 1981), which applies to both characters and RPLAs. Hence, RPLAs should exhibit personality traits that match those of the characters (Wang et al., 2024d), which could be measured via psychological scales such as the Big Five Inventory.

To evaluate RPLAs on the aforementioned dimensions, existing methodologies could be distinguished into four categories:

1. **Automatic Evaluation with Ground Truth**: Typically, datasets with ground truth are expected for evaluating character fidelity in terms of knowledge, personality and thought. While early similarity metrics such as Rouge-L (Lin, 2004) could be applied to compare RPLA responses with ground truth (Wang et al., 2024a), recent studies increasingly leverage state-of-the-art LLMs such as GPT-4 as evaluators. On one hand, evaluator LLMs can score RPLA responses based on certain criteria, or determine the superior response from two models for win rate calculation, provided with "ground truth" responses as references. However, the "ground truth" are typically synthesized by advanced LLMs (Wang et al., 2024a) . On the other hand, evaluator LLMs can be used to classify RPLA responses, and the results are then compared with ground truth labels (Wang et al., 2024d).

2. **Automatic Evaluation without Ground Truth**: As collecting ground truth data for RPLA evaluation is often challenging, several studies such as CharacterEval explore using LLMs to evaluate RPLA responses without ground truth (Shao et al., 2023; Tu et al., 2024). Instead, character profiles should be provided. This category is effective for evaluating character-independent abilities and linguistic styles, which require little knowledge about the characters. However, when it comes to characters' knowledge and thoughts, LLMs might not possess the necessary depth of relevant knowledge, especially for unfamiliar characters. This concern potentially leads to inadequately informed judgments of LLMs, and hence produces suboptimal evaluation results. Even when equipped with extensive knowledge, current LLMs still exhibit deficiencies in discerning nuanced aspects of role-playing.

3. **Multi-choice Questions**: Multi-choice questions also come with ground truth, yet they differ from "automatic evaluation with ground truth" in that they merely require RPLAs to select from a fixed set of options, rather than generating open-ended responses. This significantly reduces the output space for RPLAs, making the evaluation simpler. This method is particularly suitable for evaluating the fidelity of characters' thoughts, *e.g.*, behavior prediction (Xu et al., 2024c; Chen et al., 2024) and motivation generation (Yuan et al., 2024b; Shen et al., 2023a). For these tasks, it is impractical to require RPLAs to produce responses exactly matching the ground truth, and responses may be reasonable even if they deviate from the ground truth significantly.

4. **Human Evaluation**: Inviting human annotators to assess the performance of RPLAs is a viable and effective approach (Zhou et al., 2023a). However, it comes with several drawbacks, such as cost in terms of time and money, as well as lack of reproducibility. This method is akin to "automatic evaluation without ground truth", yet employs humans as more precise evaluators. Hence, it similarly falls short in evaluations that require in-depth knowledge about the characters, as recruiting qualified annotators who are well-acquainted with these characters can be difficult. Combining automatic evaluation with human evaluation, some efforts also fine-tune the evaluator LLMs with human annotations (Tu et al., 2024).

Previous efforts primarily focus on RPLA evaluation on widely-known established characters (Wang et al., 2024d; Ahn et al., 2024). However, it is challenging to obtain high-quality datasets and achieve precise and

nuanced evaluations for those highly complicated characters. Hence, there has recently been a growing trend in assessing LLMs' role-playing capabilities based on original characters (BosonAI, 2024; Samuel et al., 2024).

# 6 Individualized Persona(lization)

## 6.1 Definition

**Personalization** tailors LLMs to meet the unique needs, experiences, and preferences of individuals, which have been increasingly important in modern AI applications (Salemi et al., 2024). Research in this area aims at providing personalized services, adapting to the preferences of individual users or even mirroring their behaviors (Chen et al., 2023b). When such a personalized system attempts to encapsulate these aspects, it essentially engages in role-playing, emulating an individual. This process shapes **individualized persona** for RPLAs (Salemi et al., 2024), typically embodying digital clones or personal assistants for individuals.

In this paper, we categorize the applications of personalized RPLAs into three tiers, ranging from **conversation** (Gao et al., 2023b; Ahn et al., 2023) and **recommendation** (Chen et al., 2023b; Yang et al., 2023a), to autonomous agents for more complicated **task solving** (Li et al., 2024d).

1. **Conversations**: Early research for personalized RPLAs primarily focuses on personalized conversations by learning and incorporating the user persona (Cho et al., 2022; Zhou et al., 2023c; Ng et al., 2024), aligning stylistic features with user preferences to boost engagement (Zheng et al., 2021; Wang et al.). With the emergence and evolution of LLMs, personalized RPLAs become capable of handling increasingly complex and comprehensive tasks, gaining competence in complicated task-planning and tool-learning for auto-completing personalized services.

2. **Recommendation**: Conversational recommendation systems (Chen et al., 2023b; Yang et al., 2023a; Wu et al., 2023) based on LLMs have been widely regarded as the next generation of recommendation systems (Lian et al., 2024), support users in achieving recommendation-related goals through multi-turn dialogues (Jannach et al., 2021). Compared with traditional recommendations, these methods stand out with their solid foundation models, natural language interactions, and straightforward, typically nonparametric evolution (Sallam, 2023; Abbasian et al., 2023).

3. **Task Solving**: Furthermore, personalized RPLAs become increasingly competent in more complicated task solving (Yao et al., 2023a; Significant-Gravitas, 2023), such as coding (Microsoft, 2024), travel planning (Xie et al., 2024b), and research survey (Wang et al., 2024c), typically interacting with various external software. They are autonomous LLM-based agents that are deeply integrated with personal data, devices, and services (Dong et al., 2023; Li et al., 2024d). They have significantly advanced personal assistants beyond early predecessors such as Siri (Apple Inc., 2024) which struggle with complex user requests.

To build personalized RPLAs that accurately capture and portray the individualized personas, the process typically consists of two crucial steps: *1)* **Persona data collection**, which gathers the necessary data to shape the individualized personas, and *2)* **Persona modeling**, which creates models that represent these individual personas using the collected data. For persona data collection, the data can vary greatly in format, content, and modalities across different applications and tasks. We categorize this data into three types: profile, interactions, and domain knowledge, which will be detailed in §6.2. For persona modeling, the challenge is to embody the intended persona from the unprocessed persona data, which are generally massive, sparse and noisy, as will be discussed in §6.3. The evaluation of personalized RPLAs depends on specific applications, and will be discussed in §6.4.

Despite the advancement with LLMs, personalized RPLAs still face several challenges, including processing long inputs and vast search space (Chen et al., 2023b; Abbasian et al., 2023), utilizing sparsity, lengthy, and noisy user interactions data (Zhou et al., 2024c), learning domain-specific knowledge for understanding user profiles (Zhang et al., 2023c), understanding multi-modal contexts (Dong et al., 2023), ensuring privacy and ethical standards (Benary et al., 2023; Eapen & Adhithyan), and optimizing response time for seamless integration into real-time applications.

Table 3: Overview of existing role-playing datasets with individualized personas.

| Datasets | #Profile | #Interactions | Domain | Lang. | Source |
|---|---|---|---|---|---|
| PERSONA-CHAT (Zhang et al., 2018) | 1,155 | 10,907 | - | EN | Crowdsourcing |
| ConvAI (Dinan et al., 2020) | 1,155 | 17,878 | - | EN | Crowdsourcing |
| Qianyan (Baidu, 2020) | 23,000 | 23,000 | ✓ | ZH | Unknown |
| P-Ubuntu (Li et al., 2021) | 1000k | 1000k | - | EN | Ubuntu |
| P-Weibo (Li et al., 2021) | 1000k | 1000k | - | ZH | Weibo |
| FoCus (Jang et al., 2022) | 14,452 | 14,452 | ✓ | EN | Crowdsourcing |
| MPCHAT (Ahn et al., 2023) | 15,000 | 15,000 | - | EN | Reddit |
| OpinionQA (Santurkar et al., 2023) | 18,339 | 1,476 | - | EN | Crowdsourcing |
| SPC (Jandaghi et al., 2023) | 4,723 | 10,906 | - | EN | LLM |
| COMSET (Agrawal et al., 2023) | 202 | 53,903 | - | EN | GoComics |
| RealPersonaChat (Yamashita et al., 2023) | 233 | 14,000 | - | JP | Crowdsourcing |
| LiveChat (Gao et al., 2023b) | 351 | 1,332,073 | ✓ | ZH | Douyin |
| KBP (Wang et al., 2023b) | 2,477 | 2,477 | ✓ | ZH | Crowdsourcing |
| Cho et al. (2023) | 10 | 560 | - | KO | Crowdsourcing |

## 6.2 Data Collection of Individualized Persona

The individualized personas for personalized RPLAs are typically represented with three distinct types of data, including **profile**, **interactions**, and **domain knowledge**, depending on the specific applications. There have been numerous datasets with individualized personas, as outlined in Table 3, covering various languages including English (Ahn et al., 2023), Chinese (Baidu, 2020), Japanese (Yamashita et al., 2023), and Korean (Cho et al., 2023).

**Profiles** Profiles are fundamental information that explicitly describes individualized personas, which are typically well-structured. Typically, they are initially set by users, and can be continuously updated. The basic elements usually include the names, gender and ethnicity of individual users in text (Santurkar et al., 2023) Besides, profiles commonly contain natural language descriptions of individuals, describing their characteristics, such as identity, hobbies, experiences and other statements (Zhang et al., 2018; Dinan et al., 2020; Gao et al., 2023b; Li et al., 2021; Ng et al., 2024), varying based on the detailed applications. Additionally, Lee et al. (2024) propose a multi-modality persona that includes elements such as the user's appearance. For example, in live streaming applications, persona data can be composed of both basic profile information — such as an individual's age, gender, and location — and domain-specific details, namely streamer characteristics such as fan numbers and streaming style Gao et al. (2023b). Additionally, profiles can contain multi-modal information. For instance, profiles in (Ahn et al., 2023) incorporate text-image pairs, which are individuals' comments for pictures on social media.

**Interactions** The interaction data capture the dynamic evolution of individualized persona. Interactions are data generated during the use of applications that implicitly portray individualized personas, such as conversations, user preferences, and other behaviors. For example, PERSONA-CHAT (Zhang et al., 2018) and ConvAI (Dinan et al., 2020) collect two-person dialogues through crowd-sourcing, while LiveChat (Gao et al., 2023b) and MPCHAT (Ahn et al., 2023) collect multiplayer conversations from Internet sources such as live streaming and Reddit. To reduce construction costs, Jandaghi et al. (2023) uses a Generator-Critic architecture with LLMs for dialogue synthesis, and Zeng et al. (2024) first conducts automatic annotation to generate conversational data from raw sources like experiences, speeches, or writings. In addition to dialogues in natural language, Agrawal et al. (2023) and Santurkar et al. (2023) introduce comic pictures and multiple-choice questions as interactions. This kind of data could be consistently collected and systematically

organized in real-world applications, offering benefits such as convenient acquisition and dynamic evolution. Hence, it plays an important role in practical applications.

**Domain Knowledge**   Incorporating domain-specific knowledge into general language models aids in the better understanding of user profiles and interactions within specific domains. This is crucial for accurately understanding user needs and ensuring the consistency of the persona in role-playing (Wang et al., 2023b). For example, incorporating a knowledge base like Wikipedia helps to provide detailed backgrounds of named entities in dialogues as a part of the whole persona (Jang et al., 2022; Wang et al., 2023b; Baidu, 2020), which promotes LLMs to better understand user personas with enriched background knowledge of relevant entities.

### 6.3   Modeling Individualized Persona

Existing methodologies for modeling individualized persona can be roughly categorized into two types: offline learning and online learning. In offline learning, the learning process is conducted on the comprehensive dataset at regular intervals, which is also referred to as batch learning. In online learning, learning happens in real-time as new data becomes available.

**Offline Learning**   This method tailors the outputs of LLMs to reflect specific personas represented in pre-existing datasets. Parameter fine-tuning emerges as the mainstream approach for offline learning, typically based on SFT and RLHF (Mondal et al., 2024; Zheng et al., 2023c; Li et al., 2024b; Jang et al., 2023). For example, Mondal et al. (2024) proposes a two-stage approach for personalizing LLMs with profile and interaction datasets. In addition, some recent studies propose techniques with nonparametric learning for LLMs personalization. For instance, Shea & Yu (2023) introduces an offline RL framework with a persona consistency critic and variance reduction, while Weng et al. (2024) integrates embedding control vectors within the model's activation states, allowing dynamic output adjustment for diverse personality traits. These methods exhibit several deficiencies: *1)* they face a fundamental trade-off between accuracy and efficiency; *2)* they are heavily reliant on the quality of datasets; *3)* more crucially, they struggle to adapt to dynamic changes in persona data, limiting their real-world applicability.

**Online Learning**   In online learning, the personas are dynamic and continuously evolving, *i.e.*, regularly updated with incoming data, the user interactions in real-world applications. This enables personalized RPLAs to quickly adapt and stay relevant to user needs and preferences. With LLMs, effective persona learning is typically nonparametric and training-free, which only involves effective management of memory and context (Dalvi Mishra et al., 2022; Kim et al., 2024; Baek et al., 2023; Zhou et al., 2024c). For this demand, retrieval modules become indispensable, especially for LLMs with limited context window (Mysore et al., 2023; Sun et al., 2024). Moreover, methodologies for effective online learning methods consider not only natural language interactions, but also non-linguistic feedback from users (Ma et al., 2023a).

Besides nonparametric methods, fine-tuning with online interactive data is also widely applied to online persona learning, including both SFT with mini-batches from on-the-fly user stream data (Qin et al., 2024) and RLHF with real-time user feedback (Ding et al., 2023b; Bai et al., 2022a). Additionally, Shaikh et al. (2024b) use fewer than 10 demonstrations to align language model outputs with a user's demonstrated behaviors through iterative DPO (Rafailov et al., 2024) training. Nevertheless, significant challenges arise in accurately recognizing and learning the sparse persona-specific features from the noisy interaction data. Besides, the personas of real users may change over time, which poses further challenges for their effective modeling and updating. Therefore, for nonparametric methods, the effectiveness heavily relies on the mechanisms of memory management and retrieval.

### 6.4   Evaluation for LLMs and Individualized Persona

For effective personalization, AI models should focus on two key aspects: understanding and utilizing personas. Specifically, they should be able to identify unique user personas and predict their future preferences, actions, and thoughts, which serves as the preliminary to provide personalized responses that embody the individualized personas, in various environments that are increasingly comprehensive and complex. Here, we introduce the evaluation methodologies for personalized RPLAs across the three application tiers, namely:

*1)* **Conversation**, which focuses on models' understanding of the persona and replication of users' talking styles; *2)* **Recommendation**, which measures how models utilize persona information to recommend items that align with user preferences; *3)* **Task Solving**, which challenges models' capabilities in integrating user personas to accomplish their personalized tasks and demands.

**Conversation**   Early work in personalization for conversations represents an initial attempt to understand the persona. In this scenario, traditional tasks include predicting the speaker's persona elements (Gao et al., 2023b; Jang et al., 2022) based on dialogues, forecasting the next utterance by considering the context and persona profile (Humeau et al., 2019), evaluating the performance of ranking models (Gao et al., 2023b; Ahn et al., 2023), and recognizing the addressee in multiplayer conversations (Liu et al., 2022). The metrics typically focus on the evaluation of accuracy, fluency (Dinan et al., 2018), similarity (Popović, 2017; Post, 2018; Lin, 2004) between generated and original responses, recall, mean reciprocal rank (MRR) (Gao et al., 2023b; Ahn et al., 2023), and manual assessments (Liu et al., 2022; Gao et al., 2023b) of query relevance, persona entailment, and response fluency. Recently, ECHO (Ng et al., 2024) introduces the Turing test to RPLA evaluation, which engages acquaintances of the target individuals to distinguish between the persons and their RPLA counterparts.

**Recommendation**   For personalized recommendation, the evaluation focuses on LLMs' capabilities in understanding and leveraging user preferences from the interaction history for future recommendation. Traditional evaluation in this field measures LLMs' ability to understand and extract user preferences(Dai et al., 2024b; Yang et al., 2023a; Maghakian et al., 2023; Liu et al., 2023c; Mysore et al., 2023), the ability to rank (Dai et al., 2023a; Hou et al., 2024; Kang et al., 2023; Liu et al., 2023a; Bao et al., 2023; Chao et al., 2024), the ability of zero-shot and few-shot recommendation (Wang & Lim, 2023; Liu et al., 2023a), and the ability of sequential recommendation (Yang et al., 2024; Liu et al., 2023a). The evaluation metrics typically include Top-$k$ accuracy and MRR to assess the effectiveness.

**Task Solving**   Personalized RPLAs have been increasingly considered to provide personalized services for task solving. These tasks and requirements are usually user-specific, which exhibit greater diversity and complexity compared to traditional conversation or recommendation. Personalized RPLAs are expected to develop a deep understanding of user preferences and adhere to their complicated instructions to satisfy user requirements. Evaluating personalized RPLAs on these tasks involves assessing not only their ability to execute foundational tasks, but also their capacity to comprehend and cater to the nuanced requirements and preferences of individuals. The evaluation primarily focuses on several key aspects, including the models' abilities in theory of mind (Zhou et al., 2023b; Sap et al., 2023; Jin et al., 2024; Su & Bao, 2024; Rescala et al., 2024; Xu et al., 2024a), tool usage (Qin et al., 2023; Li et al., 2023h; Farn & Shin, 2023; Huang et al., 2023d; Zhuang et al., 2024; Huang et al., 2024c), and task automation (Wen et al., 2023a; Shen et al., 2023c; Gao et al., 2023a; Valmeekam et al., 2024). More broadly, existing studies have covered the models' ability to understand and predict user needs (Tan et al., 2024; Zhang et al., 2024a), handle personal data securely (Yim, 2023; Wu et al., 2024d; Kumar et al., 2024; Wu et al., 2024a; Yin et al., 2024), interact with information from external tools or apps (Yuan et al., 2024a; Huang et al., 2024b; Xie et al., 2024c; Huang et al., 2024d), and execute tasks (Dong et al., 2023; Guan et al., 2023; Mucha et al., 2024) effectively as a personal assistant.

# 7   Risks Beneath RPLA Applications

While RPLAs are increasingly deployed in real-world applications, potential concerns could result in significant problems if not addressed appropriately. This section highlights the risks associated with current RPLAs, covering the following areas: *1)* toxicity, *2)* bias, *3)* hallucination, *4)* privacy violations, and *5)* technical challenges in real-world deployment.

## 7.1   Toxicity

**Inherent Toxicity in LLMs**   Recent studies have underscored the proficiency of LLMs in generating content that is not only fluent and coherent but also potentially toxic. Previous research  (Zhang & Wan, 2023; Wen et al., 2023b) has highlighted a concerning tendency of these models to produce harmful content.

Such toxic outputs not only compromise user experience but also pose significant societal risks. It can lead to the perpetuation of harmful narratives, exacerbate social divisions, and even influence public opinion and behavior in detrimental ways.

**The RPLAs Conundrum**  The issue of toxicity becomes more pronounced in RPLA settings, where LLMs are more likely to generate toxic content, aligning with characters' behaviors that might not adhere to societal moral standards (Deshpande et al., 2023). However, creating completely safe RPLAs that are capable of general role-playing remains a challenging task. The inherent presence of toxic content in human-generated data complicates the development of a clean training corpus. Moreover, such a sanitized training corpus might compromise the model's performance, particularly its ability to generalize across various tasks, including role-playing. This limitation not only affects the model's generalization ability but also its effectiveness in scenarios that may require an understanding of roles characterized by behaviors or traits that diverge from societal moral standards.

**Strategies for Balancing Safety and Performance**  Despite these challenges, recent research proposes strategies like prompt engineering and semantic censorship as means to mitigate toxicity without altering the model's fundamental parameters (Han et al., 2022; Ahn et al., 2023). These approaches aim to balance the reduction of toxic outputs with the preservation of the model's versatility and effectiveness across a broad range of applications.

## 7.2  Bias

**Bias Manifestation in Role-Playing Scenarios**  Bias can manifest in both implicit and explicit forms. **Implicit bias** refers to the RLPAs' internal attitudes or stereotypes within RPLAs that influence understanding, actions, and decisions. **Explicit bias**, on the other hand, involves conscious beliefs and attitudes that align with the presented context or assigned roles. LLMs, despite being designed to avoid outputting stereotypes directly due to safety policies such as RLHF (Ouyang et al., 2022), may still exhibit biases, particularly under RPLA conditions: *1)* **Reasoning Bias**: This issue is compounded in scenarios where LLMs are assigned specific personas, leading to implicit biases that could affect their reasoning capabilities (*e.g.*, arithmetic problems), especially in contexts involving race, gender, religion, or occupation (Zheng et al., 2023a; Kotek et al., 2023; Cheng et al., 2023a; Naous et al., 2024). *2)* **Political Bias**: For RPLAs, LLMs are expected to maintain neutrality and avoid political positions or biases. Yet, studies have demonstrated a political inclination of RPLAs towards pro-environmental, left-libertarian views (Rutinowski et al., 2023; Hartmann et al., 2023). *3)* **Role-based Bias**: In the context of role-playing, role-based bias is one of the explicit bias. Striking the right balance between maintaining the authenticity of a character and managing bias is a critical challenge. For example, if an RPLA is created with a persona like "Hitler", agent will undoubtedly express some bias to Jews due to its role, which violates the ethical standard.

**Causes of Bias in RPLAs**  These biases are thought to originate from both the models' pre-training data and user interactions (Xue et al., 2023). Specifically, imbalances in training data significantly contribute to these biases, as the predominance of certain biases within the data could lead to their incorporation into the parametric memory of LLMs. Furthermore, Perez & Ribeiro (2022) and Branch et al. (2022) highlight that LLMs are sensitive to the user prompts, which could inadvertently steer them towards biased outputs. This problem gets worse when the models are influenced by the users' negative emotions (Coda-Forno et al., 2023).

**Strategies for Mitigating Bias**  Addressing biases in RPLAs requires a multi-faceted approach: *1)* **Data Preparation Phase**: Techniques such as data cleaning could significantly mitigate biases present in the training corpus (Linardatos et al., 2020). *2)* **Development Stage**: The implementation of neutral and fairness-aware classifiers during the post-processing phase has proven to be an effective strategy for further reducing bias (Sun et al., 2019; Zafar et al., 2017). Achieving fairness in role-playing scenarios, necessitates a delicate equilibrium, ensuring fairness for roles associated with both groups and individuals. For example, an RPLA tied to a specific demographic should consciously avoid reinforcing biases. It is imperative for these models to consistently produce unbiased outputs across all individuals within a group. Research is worth

pivoting towards these dimensions, striving to minimize biases and, in turn, forge safer and more equitable systems.

**Persona Construction Bias**   The prevailing instantiation of persona is often seen as simple and somehow superficial. Although most implementations of persona are helpful for basic character segmentation, they often overlook the deeper characteristics and complexities that shape character behavior (Chen et al., 2023c; Zhou et al., 2023a; Shao et al., 2023; Tu et al., 2024; Yuan et al., 2024b). For example, a conventional persona can contain basic demographics such as age, occupation, textual description of personality, *etc*. However, these aspects alone are insufficient to fully capture nuanced decision-making processes and behavioral patterns of a character. The current persona construction also lacks the flexibility and adaptability needed for specific scenarios influenced by unique events or individual actions. Therefore, it is crucial to refine and broaden the constructed dimension of persona to better understand and predict character behavior across various role-playing settings. By incorporating more detailed and specific attributes into personas, the comprehensiveness of character representation can be enhanced, improving the effectiveness and authenticity of interactions within role-playing environments.

### 7.3   Hallucination

**Hallucination in RPLAs**   Hallucination in LLMs refers to instances when these models produce factually incorrect information, a challenge particularly pronounced in knowledge-intensive tasks (Wang et al., 2023g). Role-playing, a task requiring a deep understanding of specific roles, is also one of the knowledge-intensive tasks. For hallucination of RPLAs, following Shao et al. (2023), we define behaviors that agents respond in ways that do not fit assigned roles as ***Character Hallucination***. For example, Shakespeare is not supposed to know anything about World War II. Such a hallucination prevails in language models and detracts from the system's overall effectiveness and reliability (Li et al., 2016; Zhang et al., 2018). In particular, there is a kind of hallucination when LLMs play a role for a specified period. Ahn et al. (2024) names this point-in-time character hallucination. For example, the agent simulating *Harry Potter* at an early age erroneously mentions a future event.

**Mitigating Hallucinations in RPLAs**   When encountering topics beyond their assigned characters, RPLAs are expected either to demonstrate ignorance or to refrain from answering, diverging from conventional solutions to hallucinations, such as incorporating external knowledge bases. Recent efforts, such as those by Shao et al. (2023), focus on adjusting the model through fine-tuning, teaching RPLAs to either forget knowledge or to explicitly express a lack of knowledge in their responses. However, this area remains relatively underexplored in the era of LLMs. Exploring alternative unlearning strategies (Neel et al., 2021; Pawelczyk et al., 2023), could also be a promising direction. These approaches may offer novel ways for RPLAs to manage out-of-scope knowledge more effectively, underscoring the importance of further investigation in this field.

### 7.4   Privacy Violations

**Privacy Challenges in LLMs**   Privacy concerns in LLMs are increasingly pressing. Even with advanced safety measures like those in OpenAI's GPT-4 (OpenAI, 2023), these models may still be susceptible to complex, multi-step attacks aimed at extracting private information, as noted by Li et al. (2023e). A further concern is the ability of LLMs to identify individuals from limited data. Sweeney (2002) highlights that many in the U.S. population could be uniquely identified using just a few attributes. Staab et al. (2023) extend this concern to LLMs, which could potentially recognize individuals based on specific details like location, gender, and birth date.

**Hidden Danger of Privacy Violations in RPLAs**   In role-playing scenarios, the potential for privacy violations represents a significant and hidden danger. The risk of inadvertently revealing personal information, such as email addresses or phone numbers, should not be understated, as it poses serious threats, including identity theft and unauthorized access to sensitive data. The practice of assigning specific individual personas to LLMs, aimed at eliciting private details, demands meticulous oversight to prevent such breaches. Ensuring

robust safeguards against these vulnerabilities is not just a technical necessity but a fundamental responsibility to protect users from the severe consequences of privacy violations.

**Strategies for Enhancing Privacy**   To tackle these privacy issues, a comprehensive strategy is necessary. Employing text anonymization tools is a key step, effectively removing personal data from interactions. Ensuring that RPLAs adhere to strict privacy protection protocols is also crucial, preventing them from engaging in or prompting conversations that might invade privacy. Another promising development is the creation of specialized tools designed to detect and prevent privacy leaks, like ProPILE (Kim et al., 2023d). As RPLAs continue to evolve, so too must the strategies for protecting user privacy. Future research should focus on refining and expanding the methods available for privacy protection, ensuring that RPLAs are used safely and responsibly. Enhancing these safeguards will be paramount for maintaining trust in LLM technologies, particularly in sensitive applications like role-playing scenarios where the risk of privacy breaches is heightened.

### 7.5   Technical Challenges in Real-world Deployment

When deploying RPLAs in real-world scenarios, several key issues arise that could significantly affect user experience and the effectiveness of these models.

**Lack of Social Intelligence and Theory of Mind**   Social intelligence and theory of mind (Premack & Woodruff, 1978), are the ability to perceive and reason about the inner world of oneself and others, which are indispensable for LLMs to simulate socially intelligent entities (Kosinski, 2023; Sap et al., 2023). However, such abilities in current LLMs remain to be improved (Shapira et al., 2023; Zhou et al., 2024a; Light et al., 2023; Kim et al., 2023c), which poses significant challenges for RPLAs concerning the following issues: *1)* **Inability to Provide Adequate Emotional Support and Values**: Social intelligence and theory of mind are essential for RPLAs to effectively provide emotional support and values to users. This involves perceiving users' emotions and interpreting their beliefs, intentions and needs. However, existing LLMs still fall short in these abilities, hindering RPLAs from offering adequate emotional support to users. *2)* **Tendency towards Ego-centric Behavior**: Rather than focusing on users' emotional needs, current RPLAs often exhibit a preference for showcasing their own personas and steering conversations towards their interests. This might limit the diversity and depth of role-playing interactions (Xu et al., 2022), as focusing excessively on agents' self-persona without adequately considering the users may detract from the realism of the conversation and degrade the user experience.

**Long-context Challenges**   When encountering extremely long context text, the limitation of max token window (Liu et al., 2023b) may also be a major obstacle to the development of RPLAs, as current LLMs struggle to robustly interpret and respond to extensive context. Specifically, this involves several key challenges: *1)* **Reasoning over Long Context**: Long context data learning requires the model to have the ability to handle long contexts and accept lengthy inputs, and more importantly, to capture long-range dependencies to integrate information and infer a more complete character persona from the massive context. *2)* **Efficiency**: In terms of computation, the high complexity of long context necessitates efficient modeling methods and approximation strategies to reduce computational overhead. *3)* **Immersion**: RPLAs need to be immersive enough to identify the truly persona-relevant parts from the sea of irrelevant information in long contexts, while also maintaining persona consistency throughout the long generated text.

**Knowledge Gaps**   In role-playing scenarios requiring detailed historical, cultural, or contextual understanding, RPLAs often exhibit gaps in knowledge. Their inability to provide in-depth and accurate domain-specific responses could lead to superficial or incorrect portrayals in complex role-playing settings. Several efforts also utilize LLMs to evaluate RPLAs for characters (Shao et al., 2023; Tu et al., 2024). Nevertheless, RPLAs may face challenges in accurately evaluating characters with which they are unfamiliar, potentially compromising the reliability of the evaluation results.

### 7.6 Anthropomorphism

While the initial motivation for creating RLPAs is positive, as they provide users with a valuable platform for expressing feelings, particularly those they may find difficult to share with other humans. There are significant differences between interacting with virtual RPLAs and real humans, and over-reliance on RPLAs can lead to several potential issues:

**Social Isolation** Frequent interaction with RPLAs might reduce the need or desire for real human contact, which could lead to the atrophy of essential social skills. Human interactions are inherently more complex, requiring nuanced understanding and empathy, which may not be fully cultivated through interactions with RPLAs. This over-reliance could result in social isolation, negatively affecting mental health by increasing feelings of loneliness, depression, and anxiety if individuals become overly dependent on virtual RPLAs.

**Manipulation of Public Opinion** RPLAs, particularly those designed or programmed without strict ethical oversight, could inadvertently or intentionally spread misinformation or rumors. This risk is especially concerning in sensitive social contexts, such as during elections, public health crises, or other situations where accurate information is crucial. For example, if RPLAs are deployed on social media and gain a large following, their influence could be substantial, especially if they do not disclose their true nature as AI systems, making it difficult for people to distinguish them from real humans. Sophisticated RPLAs could be deployed to manipulate public opinion by subtly altering the narratives they present to users. This could influence individuals' perceptions, decisions, and even voting behavior, without them realizing they are being influenced by virtual RPLAs rather than human discourse.

## 8 Closing Remarks

In this survey, we have systematically reviewed the research and applications of role-playing language agents (RPLAs), which has emerged to be a heated topic due to the success of large language models (LLMs). We categorize the personas in RPLA research and applications into three progressive types, *i.e.*, Demographic Persona, Character Persona, and Individualized Persona. This classification elucidates the developmental trajectory from generically assigned personas in RPLAs to highly personalized ones. Additionally, we have identified and enumerated various risks and ethical concerns associated with current applications of RPLAs. These issues underscore the urgent need for focused research to address and mitigate potential drawbacks in the implementation of RPLAs, making this arena still full of both research and application opportunities.

**Future Directions on RPLA Systems** From persona-assigned role-playing to personalization, the key for building RPLA systems is to reason and make decisions resembling or even transcending the roles that are given. To this end, we propose several important future directions to facilitate the construction of such RPLA applications:

1. **Causal Data Analysis for Decision-making:** Role-playing decisions must be made for justifiable reasons, necessitating models that go beyond simple mimicry of observable actions to include an understanding of their underlying causality. The complexity in extraction and confirmation of causal factors from intertwined experiences poses significant challenges that require advanced analytics and deeper data interpretation strategies to enable RPLAs to make informed and wise decisions.

2. **Improved Decision-making:** Decision-making process is not merely replicating histories, but tailored to ensure optimal outcomes for individual scenarios. This includes decisions showing advanced (if not superhuman) intelligence, avoiding mistakes, or making the best choices in tough dilemmas. Such agency requires RPLAs and the underlying LLMs to be able to comprehensively collect and utilize the context and intricacies associated with their roles.

3. **RPLA as Personal Assistants for Personal Decision-making:** The future development of RPLAs into comprehensive personal assistants signals a significant transformation. These systems could manage all facets of Internet behavior, from customized shopping and personalized travel

planning to new generation recommendation systems. By incorporating multimodal data handling, including images and videos, and linking with advanced visualization technologies, RPLAs could significantly enhance personalization and efficiency in everyday tasks.

4. **Social Simulation through Autonomous Role-Playing:** Utilizing RPLAs for social simulations can significantly extend their application by conducting elaborate experiments in diverse scenarios to study psychological and sociological behaviors. By role-playing various characters, RPLAs can serve as versatile test subjects to explore the influence of different personality traits on social intelligence, providing valuable insights into human behavior and interaction dynamics.

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

# A    RPLA Products

The recent remarkable advancements in LLMs have sparked a myriad of AI applications. Persona and personalization are central to these applications, with their demands shaping and propelling research in RPLAs. In this section, we provide a brief overview of recent trends in RPLA applications. Specifically, we distinguish RPLAs in existing products into two categories, namely **persona-oriented RPLAs** and **task-oriented RPLAs**, as listed in Table 4.

## A.1    Persona-oriented RPLA Products

Persona-oriented RPLAs typically role-play as specific characters, which has been popular in various entertainment applications, such as chatbots and game NPCs. These RPLAs are generally sourced from fictional characters, historical figures or celebrities, aligning with the research trends on character persona as introduced in §5. They are further forked for individual use cases to meet their preference. We categorize existing persona-oriented RPLA products based on their primary interaction focus, either **human-RPLA interactions** or **RPLA-RPLA interactions**.

**Interactions between Humans and RPLAs**    Persona-oriented RPLAs, such as those in Character.ai, are initially applied for human-RPLA interactions. These RPLAs can be both **initiated from established characters** and **shaped through ongoing user interactions**.

Having conversations with widely-recognized **established characters** attracts extensive interest among users. Consequently, numerous products have been developed to provide RPLAs representing these established characters, including celebrities (*e.g.*, Meta AI), historical figures (*e.g.*, Hello History) and fictional characters (*e.g.*, ChatFAI), or general individuals with specific professions or personalities (*e.g.*, Character.ai). In a more personalized manner, users can also create RPLAs with user-defined personas (*e.g.*, Character.ai, Replika). Technically, these RPLAs are typically built based on LLMs with strong role-playing capacity, with character settings briefly described in prompts. While several open-source projects and research efforts such as ChatHaruhi (Li et al., 2023a) and RoleLLM (Wang et al., 2024a) curate detailed and comprehensive character data for specific well-known characters, such practice is rarely adopted by commercial applications for generality and cost efficiency.

In many products, RPLA personas **evolve dynamically** throughout the course of interaction with users (*e.g.*, Replika, Rosebud, Rewind.ai). These RPLAs learn from and adjust to user prompts and preferences, typically with long-term memory modules. Several products aim to reproduce a "digital self" (*e.g.*, Personal.ai, Bhuman.ai). They build RPLAs to represent user personas, replicating their languages and even their physical characteristics, such as voice or visual appearance. Hence, these RPLAs support not only text chats but also video presentations and conferences, which have been adopted for sales, digital marketing, customer service, *etc*.

**Interactions among RPLAs**  Products featuring interactions among multiple RPLAs often target interactive gaming or simulations. In these scenarios, users can either act as an orchestrator of the storyline or role-play as one of the pivotal characters within the story. In Ememe.ai and AI Dungeon, users design the settings and characters of a simulation, with or without participating directly as a player, which resembles sandbox games. The characters and storylines are generated directly by one story model or based on multiple RPLAs and their interactions. In the latter case, users play as a character in the story and interact with other RPLA characters (*e.g.*, SageRPG) Furthermore, numerous products transform films, novels, and various franchises into immersive RPGs (role-playing games) with interactive RPLAs (*e.g.*, Hidden Door). Integrating RPLAs and LLMs into these games expands the possibilities for user actions and brings characters to life beyond the limitations of predefined storylines, thereby enriching the overall user experience.

## A.2 Task-oriented RPLAs

The remarkable advancements in LLMs have propelled significant development in AI applications for specialized tasks. In these applications, LLMs typically communicate in a human-like manner to foster user acceptance, and serve as domain experts providing personalized services for users, such as AI doctors and coaches. These applications are closely related to research work in the personalization of RPLAs introduced in §6. We refer to personalized agents in these products as task-oriented RPLAs. This section offers a concise overview of task-oriented RPLAs in AI products, spanning various domains, including education, healthcare, human resources, customer service, content creation, real estate, shopping, fitness, travel, and finance.

**Education**  For education, personalized agents are adopted for personalized recommendations and adaptive learning, serving both educators and learners. For learners, RPLAs can personalize the learning journey by tailoring content and recommendations to individual learning styles and paces for optimal engagement (*e.g.*, Jagoda.AI, Khan Academy's Khanmigo, Duolingo Max). For educators, RPLAs can alleviate administrative tasks by recommending personalized teaching materials and assessments, as well as creating multilingual instructional content (*e.g.*, Eduaide.Ai).

**Human Resource**  In human resources, RPLAs can provide tailored assistance for job seekers based on their profiles and interests to aid their career navigation. They offer personalized support in answering interview questions, career advice, and even customizing interview preparation materials (*e.g.*, Autonomous HR Chatbot, AI Interview Coach, Careers AI, Huru AI).

**Real Estate**  LLMs have been widely adopted for content generation and recommendation in the real estate industry. They can generate blog articles and attractive descriptions and recommend a list of potential interests for users based on their needs. By analyzing user preferences and needs, these products can generate

tailored property recommendations to enhance user experience (*e.g.*, Epique, Listingcopy). Moreover, LLMs enable these platforms to create compelling and informative content, such as property descriptions and neighborhood guides, attracting potential buyers and renters. These personalized AI products could also analyze vast amounts of market data to provide users with actionable insights and data-driven strategies about real estate.

**Content Generation**   AI products for content generation aim to assist in or even automate the production of creative and personalized content via simply natural language interactions. These products support a wide array of content types, including text, images, audio, and videos, tailored to various styles, themes, scenes, and objectives. With state-of-the-art AI models, these products push the boundaries of human creativity. HyperWrite and AI Story Generator specialize in creative textual writing, whereas DALL · E 3 and Sora are developed to create image and video content. Several products specialize in social media posts, such as EZAi AI and AI Majic. These products provide services for social media bloggers by analyzing user interactions and offering insights into audience preferences by providing keywords and detailed analysis. This analysis helps optimize content impact and strengthen the connection between bloggers and their audiences. Besides, LLMs could also role-play as assistants to aid users in grasping online content via summarization and interactive question-answering, thus fostering enhanced understanding and engagement (*e.g.*, X's Grok, Bibigpt).

**Health**   In the healthcare domain, personalized agents provide tailored medical services for patients, including general health guides, scheduling logistics, prescription information, patient care guidelines, and assistance in medical software operations for the aged. These agents are typically personalized based on patients' personal data, supported by LLMs and knowledge graphs in medical domains (*e.g.*, IBM Watson Health and Babylon Health). They could interact with patients in natural language and continuously adapt to their personalized contexts. Hence, these agents could well comprehend patients' intent, generate appropriate responses and recommendations, and continuously optimize their performance and effectiveness based on patient feedback and data. (*e.g.*, Ada Health and K Health)

**Travel**   For the tourism industry, personalized agents provide various services, including information provision, consultation, booking, cancellation, and complaint handling on social apps. On the one hand, many products offer digital concierge services (*e.g.*, HiJiffy), delivering automated services customized to suit diverse user needs, including customers' queries, accents, emotions, preferences, and other characteristics. This reflects the brand's commitment to superior service. On the other hand, travel agents are popular in many products (*e.g.*, AI Adventures, Trava). These travel agents can pinpoint users' travel needs and provide personalized services, including identifying popular destinations, grasping the underlying intentions behind user queries, and meeting customers' emotional needs. These products could refine their services and anticipate market shifts in tourism by analyzing collected user data.

**Customer Service**   In customer service, personalized agents assist to enhance problem-solving efficacy and user engagement. They offer 24/7 support across diverse domains and boost first-contact resolution rates. RPLAs leverage user feedback and implicit actions to optimize their personalization and elevate the user experience (*e.g.*, Ebi.Ai, boost.ai, Jason AI, Ada). Comprehensive AI assistants deliver and analyze user inquiries, preferences, and context to provide tailored responses. They also extract actionable insights from conversational data. For example, Viable targets businesses by empowering them with valuable understanding gleaned from large volumes of user feedback. This enables companies to make data-driven product and service improvements based on real customer needs and pain points.

**Shopping**   In the shopping industry, personalized agents simulate in-store conversations to provide tailored product recommendations, match items, and discover trends based on user preferences. Products such as Shopping Muse (*e.g.*, Dynamic Yield by Mastercard) offer relevant product suggestions and help users find items that match their style and interests based on user preferences and needs through human-agent conversations.

**Fitness**   For fitness, personalized agents enhance the fitness training experience, both at home and in the gym. RPLA aims to play the role of coaches in setting realistic goals, adapting exercises based on

progress and abilities, and providing multimodal feedback. Platforms such as WHOOP Coach and Humango collect users' biometric information, physical characteristics, and fitness levels. Through natural language conversations, these agents offer personalized training plans tailored to individual preferences and needs. By making personalized health coaching more accessible, these AI-powered RPLAs democratize access to expert guidance and support for a wider audience.

**Office** For office productivity, task-oriented RPLAs can role-play as the copilot for individual workers based on their office data, such as document files and code repositories. Hence, they deliver context-aware assistance for user requests, such as generating content and providing insights. For example, Microsoft 365 Copilot integrates various user data to deliver intelligent services that respond to user queries and enable more convenient interaction with applications. It integrates with Microsoft Graph and utilizes user data from various sources, including documents, email threads and others, with continuous learning mechanisms to improve its performance over time. Similarly, GitHub Copilot integrates individual code repositories and serves as the copilot to boost the productivity of programmers. These personalized RPLAs empower users to streamline their workflows and enhance productivity within the office environment.

Table 4: Overview of RPLA applications and products based on LLMs. For personalized data, "Personal Profile" refers to data about one's identity, including age, appearance, voice, and biographical information. "Behavior History" denotes data derived from interactions between users and applications, representing user behavior patterns. "File" pertains to documents and computer files containing private knowledge regardless of personal identity, such as code and manuals. The three types of personalized data roughly correspond to profile, interactions, and domain knowledge in §6 respectively.

| Product | Domain | Description | Target Audience | Generation Modality | Personalized Data |
|---|---|---|---|---|---|
| *Persona-oriented RPLA Products* | | | | | |
| Character.ai | Chatbots | A general AI chat app with a wide range of characters based on individuals with specific professions or personalities | ToC | Text | - |
| Meta AI Familiar Faces | Chatbots | AI characters role-playing celebrities | ToC | Text | - |
| Hello History | Chatbots | Conversation with historical figures | ToC | Text | - |
| Chatfai | Chatbots | A general AI chat app with a wide range of characters based on individuals with specific professions or personalities | ToC | Text | - |
| Replika | Chatbots | An AI companion that serves as an empathetic friend to the user | ToC | Text | Behavior History |
| Rosebud | Chatbots | An AI friend that allows users to journal their thoughts for mental health and personal growth | ToC | Text | Behavior History File |
| Rewind | Chatbots | A personalized agent that has the context of what users have seen, heard, or said on their device | ToC/ToB | Text Audio | Personal Profile Behavior History File |
| BHuman | Chatbots | AI digital clone of oneself with added modalities of face cloning and voice cloning | ToC/ToB | Text Audio Video | Personal Profile Behavior History File |
| personal.ai | Chatbots | Train one's own AI with knowledge of oneself and their own memories | ToC/ToB | Text | Personal Profile Behavior History File |
| Ememe | Games | An AI NPC sandbox that allows users to create characters and observe their life and interactions | ToC | Text | - |
| AI Dungeon | Games | A text-based adventure game where users define the characters and the setting and also participate in the game as a character | ToC | Text | - |
| Saga | Games | An interactive fiction game where one can play as a character from pre-existing Worlds and Characters from popular franchises and media | ToC | Text | - |
| Hidden Door | Games | An interactive game that allows users to play as a character in a world that is converted from the existing movie, novel, or other types of franchise | ToC | Text | - |

| Product | Domain | Description | Target Audience | Generation Modality | Personalized Data |
|---|---|---|---|---|---|
| | | *Task-oriented RPLAs* | | | |
| GPTs | All | GPTs from GPT Store are tailored versions of ChatGPT for specific tasks developed by the ChatGPT community, with categories like image generation, writing, research, programming, and education. | ToC | Text Image | - |
| Duolingo Max | Education | AI Agent that helps users to learn English better | ToC | Text | Personal Profile Behavior History |
| Jagoda.AI | Education | Personalized educational experience | ToC | Text | Personal Profile File |
| Squirrel AI | Education | Uses LLMs and AI for adaptive learning | ToC | Text | - |
| Eduaide.Ai | Education | Eduaide.ai uses AI to generate custom teaching resources and assessments based on educator input, simplifying lesson planning in multiple languages. | ToC | Text | - |
| Squirrel AI | Education | SquirrelAI employs AI to personalize learning by analyzing student performance, adjusting content, and offering tailored resources and feedback. | ToC | Text | - |
| Autonomous HR Chatbot | Human Resource | An HR chatbot that automates interviews and uses Pinecone's semantic search, powered by ChatGPT and GPT-3.5-turbo. | ToC | Text | - |
| Huru | Human Resource | Huru AI delivers personalized interview prep, featuring a Chrome Extension for actual job listing practice and a mobile app for users on the move. | ToC | Text Video | Personal Profile Behavior History |
| Careers AI | Human Resource | A platform provides career advice and planning, helping users identify and achieve their career goals. | ToC | Text | Personal Profile Behavior History |
| Epique | Real Estate | Create a blog post, write a real estate property description, draft an account activation email, and develop Instagram content about legal service pricing. | ToB | Text | - |
| PropertyPen | Real Estate | Generates property listings provides market analysis, and automates responses | ToB | Text | - |
| Listingcopy | Real Estate | AI tool for creating property listings and attractive content for real estate agents | ToB | Text | Behavior data |
| Ada | Customer Service | Ada.cx delivers personalized customer experiences across various industries, analyzing customer data like past interactions and purchases to anticipate needs and streamline interactions. | ToB | Text | Behavior History File |
| Ebi.Ai | Customer Service | AI assistant platform for business offering customer service and support | ToB | Text | - |

| Product | Domain | Description | Target Audience | Generation Modality | Personalized Data |
|---|---|---|---|---|---|
| *Task-oriented RPLAs* | | | | | |
| Jason AI | Customer Service | AI assistant for B2B sales, enhancing lead generation and sales strategies. | ToB | Text | Behavior History File |
| Aide | Customer Service | Aide enhances customer experiences by analyzing conversations for insights, automating workflows, and boosting agent efficiency with AI. | ToB | Text | - |
| Zendesk AI | Customer Service | AI customer support agents | ToB | Text | Personal Profile Behavior History File |
| Air.ai | Customer Service | An AI sales and customer service agent that can perform an actual phone call and take actions across applications | ToB | Audio | Personal Profile Behavior History File |
| boost.ai | Customer Service | Conversational AI platform for automating customer service and internal support using chat and voice chatbots. | ToB | Text Voice | Personal Profile Behavior History |
| Viable | Customer Service | Viable offers automated user feedback analysis for actionable business insights, customizing data processing to target improvements and inform strategic decisions. | ToB | Text | - |
| HyperWrite | Content Generation | An AI writing assistant that helps users in composing essays and other texts more confidently | ToC | Text | Behavior History |
| AI Story Generator | Content Generation | A tool for generating story ideas, helping writers overcome creative blocks. | ToC | Text | Behavior History |
| EZAi AI | Content Generation | An AI app for Android and IOS that helps users generate high-quality content for social media and Blogs | ToC/ToB | Text | Behavior History |
| AI Majic | Content Generation | AI that specializes in creating and managing social media content and Blogs | ToC/ToB | Text | Behavior History |
| Jasper | Content Generation | Personalized writing suggestions | ToB | Text | Personal Profile Behavior History |
| ShortlyAI | Content Generation | Personalized content generation | ToC | Text | Behavior History |
| IBM Watson Health | Health | AI Agent that provides hidden health problems with personalized plan | ToC | Text | - |
| Babylon Health | Health | LLMs process de-identified medical data with consent, personalizing healthcare through triage, diagnosis, and health predictions. | ToC | Text | - |

| Product | Domain | Description | Target Audience | Generation Modality | Personalized Data |
|---|---|---|---|---|---|
| *Task-oriented RPLAs* | | | | | |
| AI Adventures | Travel | Use LLM and external tools (API calls) to give a personalized plan on travel plans | ToC | Text | Behavior History |
| Trava | Travel | AI travel assistant that facilitates travel bookings and itinerary management. | ToC | Text | Personal Profile |
| Shopping Muse | Shopping | Shopping Muse by Mastercard offers a tailored online shopping experience, simulating in-store conversations to recommend products, match items, and discover trends based on user preferences. | ToC | Text | Personal Profile Behavior History File |
| WHOOP Coach | Fitness | Give advice and responses for fitness goals/plans uniquely tailored to users' biometric data | ToC | Text | Personal Profile Behavior History File |
| Humango | Fitness | Give customized workout plans and engaging in conversational interactions | ToC | Text | Personal Profile Behavior History File |
| Microsoft Copilot | Office | Integration into Microsoft 365 apps like Word, Excel, PowerPoint, Outlook, and Teams for enhanced creativity and productivity. | ToC | Text | Personal Profile Behavior History File |
| GitHub and features code completion developed Github Copilot | Office | Personalized AI coding copilot. | ToC | Text | Personal Profile Behavior History File |
| NexusGPT | Office | Autonomous AI employee for productivity tasks | ToB | Text | Personal Profile Behavior History File |

