# OpenReview forum: "From Persona to Personalization: A Survey on Role-Playing Language Agents"
_TMLR — Accepted by TMLR_

### Review · Reviewer_YrAL · 2024-07-26

**Summary Of Contributions:**

This is a survey for role play language agents (RPLAs), RPLAs are designed to simulate assigned personas and are made possible due to the development of large language models (LLMs). A lot of RPLA researches has been conducted on the development, evaluation and safety measurement. At the same time, RPLA applications have flourished that focuses on personal assistants, emotional companions, etc. This paper presents a comprehensive survey detailed the definition, data collection, modeling, evaluation of three types of personas: demographic persona, character persona, individualized persona. In addition, the survey discussed about risks, limitations and future of RPLAs.

**Audience:**

Yes

**Broader Impact Concerns:**

The survey already have a detailed section on risk and broader impact literature collection and discussion.

**Claims And Evidence:**

Yes

**Requested Changes:**

None.

**Strengths And Weaknesses:**

Strength:
1. Provide detailed taxonomy on RPLA types.
2. Provide summary of RPLA construction methods.
3. Comprehensive collection of RPLA datasets.
4. Provide comprehensive collection of RPLA evaluation methods.
5. Categorize and collect on research of risk of RPLAs from five perspectives: toxicity, bias, hallucination, privacy violation, technical challenges.
Weakness:
Not that I can found.

---

> ### Author Response · Authors · 2024-08-25
>
> Thank you for your encouraging feedback and for supporting our work in this rapidly evolving field.

---

### Review · Reviewer_KvjP · 2024-08-09

**Summary Of Contributions:**

This survey paper systematically summarizes the works about role-play language agents (RPLAs) and categorizes them based on the types of persona, including demographic, character, and individualized persona. Finally, they discuss the potential issues of RPLAs and illustrate some challenges to building more socially intelligent language agents.

**Audience:**

Yes

**Broader Impact Concerns:**

No concerns if the requested changes have been addressed.

**Claims And Evidence:**

Yes

**Requested Changes:**

* Uniformly discuss the role of LLM and prompting in constructing RPLAs.

* Expand the discussion of the risks of anthropomorphism.

**Strengths And Weaknesses:**

The paper touches a trending topic and comes in timely for people who want to quickly understand the landscape of RPLAs. The organization of the paper is exceptionally clear and easy to follow, which could be helpful for people who want to get a glimpse of certain domains of RPLAs. I in general like this paper and I think the paper comes promptly considering the sheer rise of RPLAs research recently (https://sotopia-lab.github.io/awesome-social-agents).

However, there are a few points that I think could be further improved:
* A major approach, often considered one of the strongest baselines, for constructing an RPLA involves prompting LLMs, regardless of the various persona types. It would be good if the authors spent some space uniformly discussing this method.

* For the risk section, the authors do not discuss the risk of anthropomorphism. Especially in the area of RPLAs, since the goal is to mimic human behaviors, such risks could exist when people project emotions into the RPLAs and blur the difference between digital twins and real-life humans.

---

> ### Author Response · Authors · 2024-08-25
> **Response to Reviewer KvjP [1/2]**
>
> Thank you for your constructive feedback! We have made revisions in the submitted paper accordingly. Our responses to your reviews are as follows:
>
> > Q1: A major approach, often considered one of the strongest baselines, for constructing an RPLA involves prompting LLMs, regardless of the various persona types. It would be good if the authors spent some space uniformly discussing this method.
>
> Thanks for your valuable suggestions! We have added our discussion and categorization on prompting for RPLAs, as shown in Section 3.2. The prompts for RPLAs primarily consist of **persona data** that represent the intended personas, including  **descriptions** and **demonstrations**:
> - Persona descriptions (or persona profiles) represent their basic information, typically including names, backgrounds, experiences, personalities, tones, catchphrases, and various other attributes such as identity, gender, age, and relationships.
> - Persona demonstrations illustrate the representative behaviors to further align RPLAs with the intended personas. The demonstrations can be represented in many forms, such as dialogues, behaviors/interactions, preferences, stories or other modalities.
>
> For the construction of persona data to prompt RPLAs, the following methods are typically adopted:
> **Online Resource Collection** utilizes existing information from online encyclopedia like Wikipedia and Baidu Baike for widely-known characters [4][5].
> **Human Annotation** employs prefessional annotators or character fans to summarize persona descriptions [6] or to engage in role-playing conversations to gather high-quality dialogues [2].
> **Automatic Extraction** utilizes LLMs automatically extract character dialogues and other relevant information directly from original materials such as books or scripts [1][3][7].
> **Dialogue Synthesis** leverages advanced LLMs to create and expand role-playing conversation datasets. LLMs generate additional dialogues through in-context learning [1] or by role-playing as characters [3]. If provided with corresponding literature for reference (for character personas), this is close to automatic extraction and the synthesized dialogues are more faithful to the origins. Otherwise, the quality of the synthesized data is limited and often requires further filtering.
>
> Additionally, **role-playing instrucitons or requirements** could be incorporated to encourage or restrict specific behaviors of RPLAs.
>
> We have incorporated this expanded discussion into our revised manuscript in Section 3.2 to provide our readers with a more comprehensive understanding of prompting techniques for RPLAs.
>
>
> [1] Li, Cheng, et al. "Chatharuhi: Reviving anime character in reality via large language model." arXiv preprint arXiv:2308.09597 (2023).
>
> [2] Zhou, Jinfeng, et al. "Characterglm: Customizing chinese conversational ai characters with large language models." arXiv preprint arXiv:2311.16832 (2023).
>
> [3] Wang, Zekun Moore, et al. "Rolellm: Benchmarking, eliciting, and enhancing role-playing abilities of large language models." arXiv preprint arXiv:2310.00746 (2023).
>
> [4] Shao, Yunfan, et al. "Character-llm: A trainable agent for role-playing." arXiv preprint arXiv:2310.10158 (2023).
>
> [5] Tu, Quan, et al. "Charactereval: A chinese benchmark for role-playing conversational agent evaluation." arXiv preprint arXiv:2401.01275 (2024).
>
> [6] Chen, Nuo et al. “Large Language Models Meet Harry Potter: A Dataset for Aligning Dialogue Agents with Characters.” Conference on Empirical Methods in Natural Language Processing (2023).
>
> [7] Zhao, Runcong, et al. "Narrativeplay: Interactive narrative understanding." arXiv preprint arXiv:2310.01459 (2023).

---

> ### Author Response · Authors · 2024-08-25
> **Response to Reviewer KvjP [2/2]**
>
> > Q2: Expand the discussion of the risks of anthropomorphism.
>
> Thank you for your advice regarding the discussion on anthropomorphism. We have added this discussion in Section 7.6.
>
> We believe that the initial motivation for creating RPLAs is positive, as they provide users with a valuable platform for expressing feelings, particularly those they may find difficult to share with other humans. However, we recognize that there are significant differences between interacting with RPLAs and real humans, and over-reliance on RPLAs can lead to several potential issues. We have summarized these risks as follows:
>
> - Social Isolation: Frequent interaction with RPLAs might reduce the need or desire for real human contact, which could lead to the atrophy of essential social skills. Human interactions are inherently more complex, requiring nuanced understanding and empathy, which may not be fully cultivated through interactions with RPLAs. This over-reliance could result in social isolation, negatively affecting mental health by increasing feelings of loneliness, depression, and anxiety if individuals become overly dependent on virtual RPLAs.
> - Manipulation of Public Opinion: RPLAs, particularly those designed or programmed without strict ethical oversight, could inadvertently or intentionally spread misinformation or rumors. This risk is especially concerning in sensitive social contexts, such as during elections, public health crises, or other situations where accurate information is crucial. For example, if RPLAs are deployed on social media and gain a large following, their influence could be substantial, especially if they do not disclose their true nature as AI systems, making it difficult for people to distinguish them from real humans. Sophisticated RPLAs could be deployed to manipulate public opinion by subtly altering the narratives they present to users. This could influence individuals’ perceptions, decisions, and even voting behavior, without them realizing they are being influenced by AI rather than human discourse.

---

> > ### Comment · Reviewer_KvjP · 2024-08-28
> >
> > Thanks for the response!

---

### Review · Reviewer_GCmX · 2024-08-13

**Summary Of Contributions:**

This paper presents a survey of role-playing languages agent (RPLAs), which simulate different persona. In increasing levels of specificity, these are broken down as demographic, character and individualized persona. Demographic persona cover traits or descriptions shared across a group of people, which may be used for improved task-solving or simulating social interactions. Character persona represent known people or fictional characters, with a mostly well-established behavior. Individualized persona correspond to systems that adapt to user preferences and behavior, and may change over time. The paper also describes risks associated to RPLAs.

**Audience:**

Yes

**Broader Impact Concerns:**

No major concerns. There is a fairly long section about risks. I also mentioned other potential risks in the requested changes.

**Claims And Evidence:**

Yes

**Requested Changes:**

**Important**

Could you clarify how reinforcement learning methods (e.g. RLHF) fit within the construction of RPLAs? They are not specifically mentioned in table 1, and RLHF is only explicitly discussed for individualized persona. Are reinforcement learning methods applied, or can they be applied, to create RPLAs with the other persona types?

**Would strengthen paper**

If humans start interacting frequently with RPLAs instead of other humans, are there some additional risks due to the lack of (or reduced amount of) real human interactions?

As RPLAs improve, are there risks about RPLAs impersonating humans without disclosing that they are in fact AI systems?

Character persona are defined as mostly static. However, the same character may behave differently at different ages. To your knowledge, is this addressed in the literature? Character evolution over time could also potentially serve as a bridge between character and individualized persona.

**Other questions**

p6 (figure 2) and full paper: Character and individualized persona are organized by data source, construction and evaluation. Why do demographic persona follow a different structure?

p7: What does the "attractiveness" of a RPLA refer to?

p9: Do the selfish agents contribute more to the collective good than fair ones? How do these findings relate to the prisoner dilemma?

p14: For evaluation, are there approaches that combine automated metrics and human evaluation, for example by using a LLM as a tool to the annotators?

p18-19: Does bias refer to implicit biases? Otherwise, you could argue that giving a model a persona is giving it specific biases. More generally, you may want to ensure that bias is clearly defined.

**Typos/minor errors**

p7: believes -> beliefs

p10: The FAIR citation doesn't show the year.

p10: "Additionally, some studies investigate if language models can forecast characters’ future actions based
on 2) Personality": Missing segment between "on" and "2)"?

p13: "Following prior work, They": Who is "They", and remove capital t.

p16: "This method tailor"

p18: I find "There are mainly several primary aspects for such evaluation" a bit hard to understand

**Strengths And Weaknesses:**

To my knowledge, the paper provides a fairly comprehensive overview of role-playing language agents (RPLA) at this time. The organization of the paper by different persona type is sensible. There is also a fairly detailed sections about risks (although see requested changes for other potential risks).

The usage of reinforcement learning is not discussed in depth for building RPLAs (except briefly for individualized persona), while it is a common step to build large foundation models. It is unclear to me if such methods are rarely used for building RPLAs (and if so why), or not covered in detail by the paper.

---

> ### Author Response · Authors · 2024-08-25
> **Response to Reviewer GCmX [1/2]**
>
> Thank you for your detailed and constructive feedback! We have made revisions in the submitted paper, and please find our response below.
>
> > Requested Change 1 (Important): Could you clarify how reinforcement learning methods (e.g. RLHF) fit within the construction of RPLAs?
>
> Thanks for this valuable feedback. Reinforcement learning methods have been applied to RPLAs in recent months, which we summarize as follows:
>
> 1. Alignment with general users: RLHF has been commonly adopted in the industry to **improve attractiveness or mitigate certain behaviors (e.g. harmful content)**. Developers of RPLA applications typically collect preference data via inviting human annotators or gathering feedback from application users.
>
> 2. Improving social reasoning skills: RL methods could improve RPLAs in gaming or debating scenarios, such as word guessing games [1] or goal-driven conversations [2]. A significant challenge in these studies is that rewards are typically given only when the games end. Hence, some works (e.g. Sotopia-pi[2]) opt to reinforce LLMs via SFT on successful trajectories from multi-agent gaming, instead of employing RL algorithms.
>
> 3. Alignment with individual users: As we have discussed in Section 6 of this survey.
>
> Considering the practical and promising impact of RL methods on RPLAs, we have incorporated relevant content into Section 3.2 (RPLA Construction, including Table 1) of our paper.
>
> > Requested Change 2 (Would strengthen paper): Further discussions about risks about RPLA systems
>
> Thank you for your insightful comments.
>
> We believe that the initial motivation for creating RPLAs is positive, as they provide users with a valuable platform for expressing feelings, particularly those they may find difficult to share with other humans. However, we recognize that there are significant differences between interacting with RPLAs and real humans, and over-reliance on RPLAs can lead to several potential issues. We have summarized these risks as follows:
>
> - Social Isolation: Frequent interaction with RPLAs might reduce the need or desire for real human contact, which could lead to the atrophy of essential social skills. Human interactions are inherently more complex, requiring nuanced understanding and empathy, which may not be fully cultivated through interactions with RPLAs. This over-reliance could result in social isolation, negatively affecting mental health by increasing feelings of loneliness, depression, and anxiety if individuals become overly dependent on virtual RPLAs.
>
>
> - Manipulation of Public Opinion: RPLAs, particularly those designed or programmed without strict ethical oversight, could inadvertently or intentionally spread misinformation or rumors. This risk is especially concerning in sensitive social contexts, such as during elections, public health crises, or other situations where accurate information is crucial. For example, if RPLAs are deployed on social media and gain a large following, their influence could be substantial, especially if they do not disclose their true nature as AI systems, making it difficult for people to distinguish them from real humans. Sophisticated RPLAs could be deployed to manipulate public opinion by subtly altering the narratives they present to users. This could influence individuals’ perceptions, decisions, and even voting behavior, without them realizing they are being influenced by AI rather than human discourse.
>
> We have added this discussion in Section 7.6.
>
> > Requested Change 3 (Would strengthen paper):  Discussion on character evolution over time
>
> This issue has not been addressed. Most papers mentioned in the literature collect information about characters as entirely as possible and do not deal with point-in-time knowledge. However, we also find papers discussing point-in-time role-playing. TimeChara[1] reveals significant hallucination issues in current LLMs and mitigates the issue by designing auxiliary temporal and spatial tasks.
>
> In our manuscript, we have incorporated discussion on this issue in Section 5.2 and 7.3.
>
> [1] Ahn J, Lee T, Lim J, et al. TimeChara: Evaluating Point-in-Time Character Hallucination of Role-Playing Large Language Models[J]. arXiv preprint arXiv:2405.18027, 2024.

---

> ### Author Response · Authors · 2024-08-25
> **Response to Reviewer GCmX [2/2]**
>
> > Q1: p6 (figure 2) and full paper: Why do demographic persona follow a different structure?
>
> This is because the role-playing of demographic personas primarily originates from the internal knowledge of the model, without complex data sources and construction. Therefore, this section focuses more on the analysis of demographic personas, including answering whether the LLM itself has demographics and how to induce such demographic personas.
>
> > Q2: p7: What does the "attractiveness" of a RPLA refer to?
>
> The "attractiveness" reflects the extent to which an RPLA can engage users for long-term interaction in practical applications. Typically, this requires RPLAs to provide emotional value to entertain users, or generate interesting content that stimulates user imagination. Practically, this could be measured by metrics such as the number of dialogue turns and visit frequencies of the RPLA, or user engagement rates.
>
>
> > Q3: p9: Do the selfish agents contribute more to the collective good than fair ones? How do these findings relate to the prisoner dilemma?
>
> Persona has an impact on cooperative and competitive games. [1] find that the “cooperative” role significantly enhances model performance in games. [2] find the greedy/saboteur agents’ actions affected the negotiation.
>
> In the Prisoner's Dilemma, selfish agents typically lead to both parties betraying each other, resulting in a suboptimal outcome for both. Conversely, cooperative persona can lead to a better collective outcome involving trust and consideration for mutual benefits.
>
>
> [1] Huang J, Li E J, Lam M H, et al. How Far Are We on the Decision-Making of LLMs? Evaluating LLMs' Gaming Ability in Multi-Agent Environments[J]. arXiv preprint arXiv:2403.11807, 2024.
> [2] Abdelnabi S, Gomaa A, Sivaprasad S, et al. Llm-deliberation: Evaluating llms with interactive multi-agent negotiation games[J]. arXiv preprint arXiv:2309.17234, 2023.
>
>
> > Q4: p14: For evaluation, are there approaches that combine automated metrics and human evaluation?
>
> Yes, some works combine automated metrics and human annotation to ensure efficiency and effectiveness.
> For example, [3] use human annotations to train a Reward Model for scoring.
>
> [3] Tu Q, Fan S, Tian Z, et al. Charactereval: A chinese benchmark for role-playing conversational agent evaluation[J]. arXiv preprint arXiv:2401.01275, 2024.
>
>
>
> > Q5: p18-19: Does bias refer to implicit biases?
>
> Good point! We acknowledge that defining bias in role-playing scenarios is a necessary task.
>
> Bias can manifest in both implicit and explicit forms.
> - Implicit bias refers to the RLPAs' internal attitudes or stereotypes within RPLAs that influence understanding, actions, and decisions.
> - Explicit bias, on the other hand, involves conscious beliefs and attitudes shaped by the presented context or assigned roles.
>
> In the context of role-playing, role-based bias presents a particularly challenging issue. For example, if an RPLA is created with a persona like ‘Hitler who is respectful towards Jews,’ the character still carries the identity and historical context of ‘Hitler.’ This raises the question of whether a persona can ever be fully separated from its inherent biases. Striking the right balance between maintaining the authenticity of a character and managing bias is a critical challenge.
>
> We have added detalied discussion in Sec 7.2.

---

> > ### Comment · Reviewer_GCmX · 2024-09-11
> >
> > Thank you for your response and the changes you made to the paper.

---

### Author Response · Authors · 2024-08-25
**General Response from Authors**

We sincerely thank all the reviewers for their comments and feedbacks. We have responded to all of the questions, and made corresponding revisions in the submitted paper, including adding necessary discussions, fixing typos, adding new papers from the community, and clarifying some statements. Revisions in the PDF are highlighted in purple.

---

### Decision · Action_Editor_qiew · 2024-09-21

**Recommendation:** Accept as is

**Comment:**

All reviewers are in favor of acceptance. Additionally, the authors have already updated the manuscript to reflect the limited concerns of the reviewers.

**Audience:**

Researchers interested in simulated social interactions or agents - human interactions.

**Claims And Evidence:**

This is a survey summarizing the state of role-playing language agents including potential challenges, safety concerns, and application domains. Agent specifications include individualized details (e.g. demographic).  Reviewers felt the work to be timely and easy to follow.